# Translational control by DHX36 binding to 5′UTR G-quadruplex is essential for muscle stem-cell regenerative functions

Xiaona Chen[1,12], Jie Yuan[2,12], Guang Xue[1], Silvia Campanario [3,4], Di Wang[5,6], Wen Wang[7], Xi Mou [8], Shiau Wei Liew[8], Mubarak Ishaq Umar[8], Joan Isern[3,4], Yu Zhao [1], Liangqiang He[2], Yuying Li[1], Christopher J. Mann[3], Xiaohua Yu[5], Lei Wang[5,9], Eusebio Perdiguero [3], Wei Chen [7], Yuanchao Xue [5,6], Yoshikuni Nagamine[10], Chun Kit Kwok[8,11], Hao Sun [2,13✉], Pura Muñoz-Cánoves [3,4,13✉] & Huating Wang [1,13✉]

Skeletal muscle has a remarkable ability to regenerate owing to its resident stem cells (also called satellite cells, SCs). SCs are normally quiescent; when stimulated by damage, they activate and expand to form new fibers. The mechanisms underlying SC proliferative progression remain poorly understood. Here we show that DHX36, a helicase that unwinds RNA G-quadruplex (rG4) structures, is essential for muscle regeneration by regulating SC expansion. DHX36 (initially named RHAU) is barely expressed at quiescence but is highly induced during SC activation and proliferation. Inducible deletion of *Dhx36* in adult SCs causes defective proliferation and muscle regeneration after damage. System-wide mapping in proliferating SCs reveals DHX36 binding predominantly to rG4 structures at various regions of mRNAs, while integrated polysome profiling shows that DHX36 promotes mRNA translation via 5′-untranslated region (UTR) rG4 binding. Furthermore, we demonstrate that DHX36 specifically regulates the translation of *Gnai2* mRNA by unwinding its 5′ UTR rG4 structures and identify GNAI2 as a downstream effector of DHX36 for SC expansion. Altogether, our findings uncover DHX36 as an indispensable post-transcriptional regulator of SC function and muscle regeneration acting through binding and unwinding rG4 structures at 5′ UTR of target mRNAs.

[1] Department of Orthopaedics and Traumatology, Li Ka Shing Institute of Health Sciences, Chinese University of Hong Kong, Hong Kong SAR, China. [2] Department of Chemical Pathology, Li Ka Shing Institute of Health Sciences, Chinese University of Hong Kong, Hong Kong SAR, China. [3] Department of Experimental & Health Sciences, Universitat Pompeu Fabra (UPF), CIBERNED, ICREA, Barcelona, Spain. [4] Centro Nacional de Investigaciones Cardiovasculares (CNIC), Madrid, Spain. [5] Key Laboratory of RNA Biology, Institute of Biophysics, Chinese Academy of Sciences, Beijing, China. [6] University of Chinese Academy of Sciences, Beijing, China. [7] Department of Biology, Southern University of Science and Technology, Shenzhen, China. [8] Department of Chemistry and State Key Laboratory of Marine Pollution, City University of Hong Kong, Kowloon Tong, Hong Kong SAR, China. [9] College of Life Sciences, Xinyang Normal University, Xinyang, China. [10] Friedrich Miescher Institute for Biomedical Research, Novartis Research Foundation, Basel, Switzerland. [11] Shenzhen Research Institute of City University of Hong Kong, Shenzhen, China. [12] These authors contributed equally: Xiaona Chen, Jie Yuan. [13] These authors jointly supervised this work: Hao Sun, Pura Muñoz-Cánoves, Huating Wang. ✉email: haosun@cuhk.edu.hk; pura.munoz@upf.edu; huating.wang@cuhk.edu.hk

Skeletal muscle tissue homeostasis and regeneration rely on muscle stem cells, also known as satellite cells (SCs), which reside in a niche beneath the basal lamina attached to the myofibers. SCs are uniquely labeled by the expression of the paired box (Pax) transcription factor Pax7[1–4]. They are typically in quiescence, a state of prolonged and reversible cell cycle arrest. Upon activation caused by injury or disease, SCs quickly re-enter the cell cycle and proliferate as myoblasts, a process orchestrated by the rapid induction of the master transcription factor (TF) MyoD. Subsequently, most myoblasts express Myogenin and differentiate, while a subset undergoes self-renewal to restore the quiescent SC pool[1,2]. Deregulated SC activity contributes to the progression of many muscle-associated diseases[5]. At the cellular level, every phase of SC activity is tightly regulated by both intrinsic and niche-derived extrinsic factors[2,6]. Elucidation of factors and molecular regulatory mechanisms governing SC function thus is very important, being the first step toward successfully using these cells in therapeutic strategies for muscle diseases.

RNA-binding proteins (RBPs) regulate all aspects of RNA metabolism, including splicing and processing of mRNA-precursors (pre-mRNAs) in the nucleus, the exporting and localizing of mRNAs to distinct subcellular regions in the cytoplasm and mRNA translation and degradation[7]. Post-transcriptional regulation of gene expression by RBPs allows cells to orchestrate rapid changes in RNA or protein levels without altering transcription. Recent evidence suggests the contribution of post-transcriptional regulation to SC activities. For example, MyoD transcripts accumulate in quiescent SCs (QSCs), allowing rapid MyoD protein production as cells activate. High expression of Staufen1 in QSCs prevents MyoD mRNA translation by interacting with the secondary structure formed at its 3′ UTR[8]. Additional studies showed that RBP-mediated RNA degradation played key role in SCs. In fact, proteins binding to AU-rich elements (ARE) located in the 3′ UTRs of many mRNAs, such as AUF1[9], TTP[10], and HuR[11], regulate SC quiescence maintenance, activation, and differentiation through modulating the stability of their interacting mRNAs. Still, molecular insight into SC post-transcriptional regulation remains largely unknown.

DHX36, a DEAH-Box RNA and DNA helicase (also known as RHAU or G4R1)[12,13], has emerged as a key RBP/helicase capable of binding and unwinding G-quadruplex (G4) structures, which are formed by guanine-rich nucleic acids harboring the motif $[G_X–N_{1–7}–G_X–N_{1–7}–G_X–N_{1–7}–G_X]$, whereby x is 3–6 nucleotides (nt) and N corresponds to any nt. These G4 structures typically consist of four tracts of guanines arranged in parallel or anti-parallel strands that align in stacked tetra planes, which are stabilized by Hoogsteen hydrogen bonds and a monovalent cation. G4 structures can be found in both DNAs and RNAs and provide additional layers of transcriptional or post-transcriptional regulation. RNA G4 (rG4) structures, in particular, are thought to participate in post-transcriptional regulation of mRNAs, including pre-mRNA processing and mRNA turnover, targeting, and translation[14–16], although the precise mechanisms are poorly understood. To date, only a handful of proteins, including DHX36, have been shown to bind and unwind rG4 in vitro. DHX36 binds both DNA and RNA G4 structures with high affinity and specificity via a conserved N-terminal region known as the RHAU-specific motif (RSM)[17,18]. DHX36 can also promote mRNA translation by unwinding rG4s formed at 5′ UTRs. For example, during cardiac development, DHX36 binds to and unwinds the rG4 structure formed at the 5′ UTR region of Nkx2-5 mRNA, which is a key TF for heart development, thereby promoting its translation[19]. DHX36 can also exert post-transcriptional regulatory roles at other levels. In fact, DHX36 was initially named RHAU (RNA helicase associated with AU-

rich element) and shown to facilitate mRNA deadenylation and decay through direct association with the ARE element at 3′ UTR of uPA mRNA, which leads to recruitment of PARN deadenylase and exosome[12]. Moreover, DHX36 promotes the maturation of miR-26a by binding and unwinding the rG4 formed in pre-miR-26a[20], and it can also resolve the rG4 formed in p53 pre-mRNA, which is necessary for maintaining the 3′-end processing following UV-induced DNA damage[21]. Very recently, cross-linking immunoprecipitation sequencing (CLIP-seq) was used to profile DHX36 targets in HEK293 cells overexpressing DHX36, revealing that DHX36 preferentially interacts with G-rich and G4-forming sequences, which increases the translational efficiency of its target mRNAs[22]. However, as this study was based on DHX36 overexpression, it is not clear whether the endogenous DHX36 protein exhibits similar binding dynamics, or whether such a binding profile also exists in other cells. DHX36 has a broad tissue expression and is indispensable for normal development. Accordingly, ablation of Dhx36 in mouse is embryonically lethal and tissue-specific knockout of Dhx36 shows that it is required for hematopoiesis, spermatogonia differentiation, and cardiac development[19,23,24]. However, whether DHX36 has a regulatory role in the mRNA metabolism of somatic stem cells has never been addressed.

In this work, we investigate the role of DHX36 in SC regenerative functions. By specifically inactivating Dhx36 in mouse muscle SCs, we find that DHX36 is required for normal muscle development and regeneration in adults. Of the distinct myogenic stages, SC proliferation is particularly attenuated upon Dhx36 deletion. Mechanistically, CLIP-seq shows that endogenous DHX36 binds to a large number of sites on thousands of mRNAs; these binding sites are G-rich and have a high potential to form rG4 structures. Subsequent polysome profiling reveals that Dhx36 loss leads to decreased translational efficiency of the target mRNAs, with DHX36 binding to their 5′ UTR G4 sites, suggesting that the DHX36 5′ UTR rG4 interaction functions to facilitate mRNA translation. Among the DHX36-associated targets, we further confirm the formation of rG4 at the 5′ UTR of Gnai2 (guanine nucleotide-binding protein G(i) subunit alpha-2), with DHX36 binding causing rG4 unwinding to facilitate Gnai2 translation during SC activation and proliferation, thus identifying GNAI2 as a downstream effector of DHX36 in regulating SC activity during muscle regeneration. Further in-depth analyses of the integrated CLIP-seq and polysome profiling datasets also reveal previously unknown aspects of DHX36 binding and post-transcriptional regulation of mRNA processing. Altogether, our findings uncover the indispensable function of DHX36 in muscle stem cells during adult muscle regeneration and provide a comprehensive mechanistic understanding of how this RNA helicase orchestrates post-transcriptional processes.

## Results

**Activation-induced DHX36 is required for muscle formation.** To dissect whether DHX36 functions in SCs and muscle regeneration, we first examined Dhx36 expression dynamics during SC myogenic progression. Quiescent SCs were isolated by fluorescence-activated cell sorting (FACS) from Pax7-nGFP mice[25], either fixed in situ by 0.5% paraformaldehyde (PFA) prior to muscle digestion ($SC_{T0}$) or from muscles digested without fixation ($SC_{T8}$) (representing partially activated SCs by the 8-h isolation process)[26]. FACS-isolated $SC_{T8}$ were further cultured for 24, 48, or 72 h, giving fully activated and proliferating SCs (Fig. 1a and Supplementary Fig. 1a). RNA-sequencing (RNA-seq) analysis of these cells revealed that the mRNA level of Dhx36 was very low in $SC_{T0}$ and started to increase in $SC_{T8}$, with a peak at 48 h, and remained high at 72 h (Fig. 1b); this suggests that

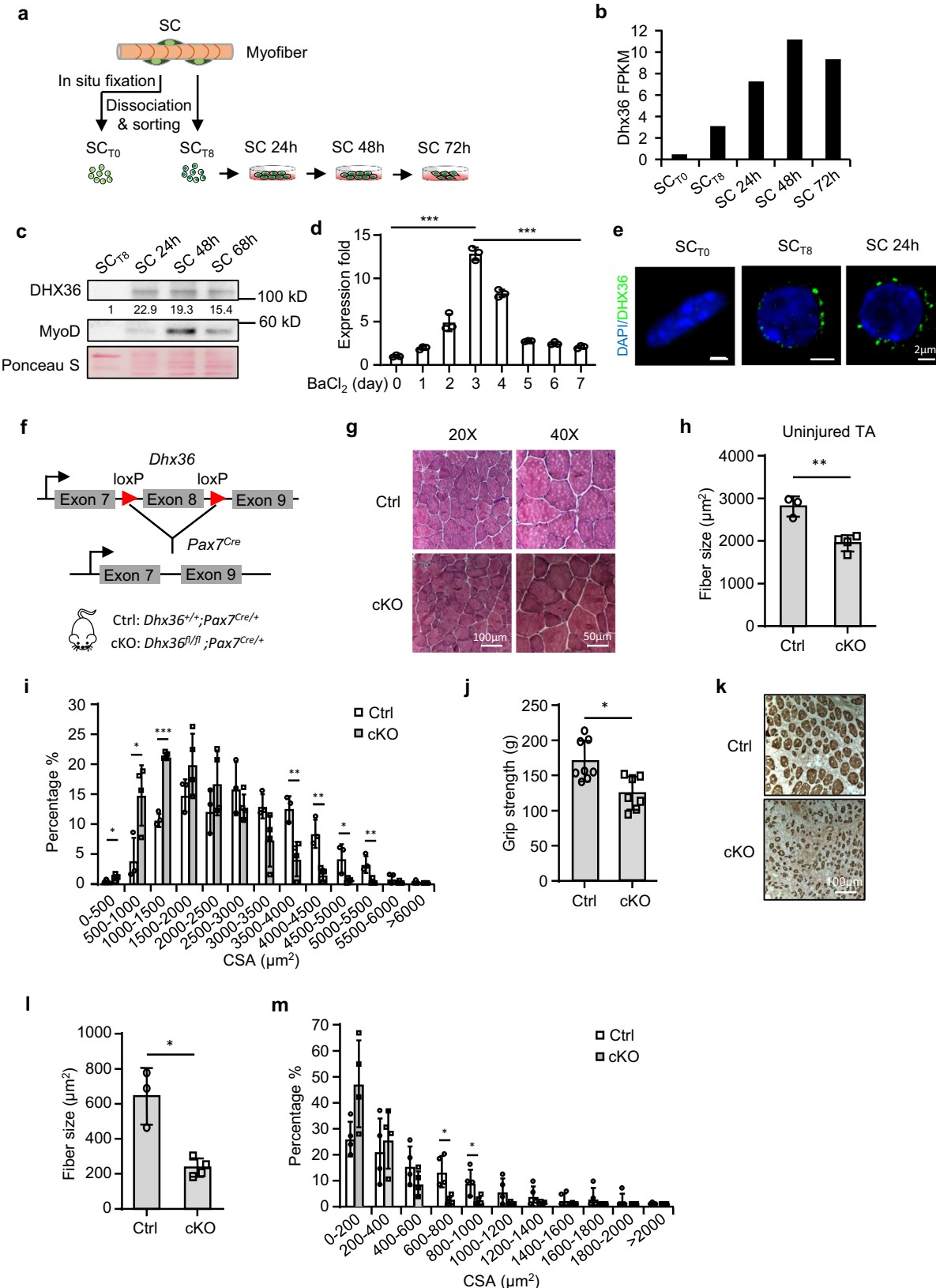

*Dhx36* expression is concomitant with the full activation and proliferation of SCs (Supplementary Fig. 1b). Western blot analysis revealed that DHX36 protein was evidently induced in SCs cultured for 24 h, as its level at $SC_{T8}$ was nearly undetectable; this high level of expression remained at 48 and 68 h (Fig. 1c). To further confirm *Dhx36* expression dynamics in SCs in vivo, muscles of C57BL/6 mice were injected with barium chloride ($BaCl_2$), which induces acute damage and regeneration. SCs can rapidly activate after muscle injury, reaching a peak of proliferation at 3 days post injury (dpi), and they are mostly differentiated by 7 dpi, coinciding with the initiation of myofiber repair[27,28]. Indeed, *Dhx36* mRNAs were highly induced at 3 dpi

**Fig. 1 Activation-induced DHX36 is required for muscle formation. a** Schematic for isolation of quiescent satellite cells (SC$_{T0}$) after in situ fixation, freshly isolated SCs without prior fixation (SC$_{T8}$) and SCs cultured for 24, 48, or 72 h. **b** RNAs from the above cells were subject to RNA-seq and FPKM of *Dhx36* mRNA is plotted. **c** DHX36 protein in the above SCs were examined by western blot. The relative intensity of each band is calculated by normalizing with total proteins stained by Ponceau S. The intensity for SC$_{T8}$ was set as 1. **d** Mouse TA muscles were injected with barium chloride (BaCl$_2$) and the muscle homogenates were collected at the designated dpi for qRT-PCR measurement of *Dhx36* mRNAs. *n* = 3 independent experiments; *P* = 0.0000097 and 0.000015. **e** Immunofluorescence staining of DHX36 protein in SC$_{T0}$, SC$_{T8}$ or SC 24 h. Scale bar: 2 μm. **f** Breeding strategy to generate *Dhx36* cKO mouse. **g** H&E staining of TA muscle from adult Ctrl or *Dhx36* cKO mice. Scale bar = 100 μm (×20 images) or 50 μm (×40 images). **h** Average fiber size in the above muscles was measured, Ctrl: *n* = 3 mice; cKO: *n* = 4 mice; *P* = 0.003. **i** Cross-section area (CSA) of each fiber from the above muscles (Ctrl in white and cKO in gray) was quantified and the distribution is shown. From left to right, *P* = 0.014, 0.027, 0.000013, 0.008, 0.003, 0.026, 0.007. **j** Grip strength of fore and hindlimb muscles were measured in Ctrl and cKO mice, Ctrl: *n* = 8 mice; cKO: *n* = 7 mice; *P* = 0.023. **k** Immunohistochemistry staining of eMyHC was performed on sections of TA muscles from Ctrl and cKO mice on 7 dpi, scale bar = 100 μm. **l** The average fiber size of newly formed fibers was quantified from the above eMyHC+ fibers, Ctrl: *n* = 3 mice; cKO: *n* = 4 mice; *P* = 0.022. **m** Distribution of CSA of the above fibers (Ctrl in white and cKO in gray). *P* = 0.011 and 0.033. Data are represented as mean ± s.d., Student's *t* test (two-tailed unpaired) was used to calculate the statistical significance (**d**, **h**, **i**, **j**, **l**, **m**): \**P* < 0.05, \*\**P* < 0.01, \*\*\**P* < 0.001. Source data are provided as a Source Data file.

(6.6-fold increase as compared to uninjured) and decreased by 8.8-fold at 7 dpi (Fig. 1d). In addition, in C2C12 mouse myoblast cells cultured in differentiating medium (DM), the DHX36 protein level also decreased during the differentiation course, particularly at late differentiation stages (72 and 120 h), further confirming that it was downregulated during myogenic differentiation (Supplementary Fig. 1c). Thus, Dhx36 expression is induced in activating and proliferating SCs. Lastly, when examining its subcellular localization, we found that DHX36 protein was exclusively present in the cytoplasm of activated SCs and was associated with visible puncta (Fig. 1e), suggesting that it mostly functions as a rG4 rather than a DNA G4 regulator.

To investigate the role of DHX36 in skeletal muscle formation and regeneration, we first assessed whether constitutive deletion of *Dhx36* affects muscle formation. We conditionally deleted *Dhx36* in the Pax7-expressing embryonic myogenic precursor cells by crossing *Pax7^{Cre}* mice with *Dhx36^{fl/fl}* mice in which the exon 8 of *Dhx36* was flanked by loxP sites[23,29,30] to generate conditional *Dhx36* knockout (cKO) mice (Fig. 1f). cKO mice showed no visible muscle developmental defects and had a normal muscle/body weight ratio, but they did have a slightly lower adult body weight (with a small yet significant difference), as compared to control (Ctrl) mice (Supplementary Fig. 1d, e). Further, as compared to Ctrl mice, *Dhx36* cKO mice had a 30.7% decreased average fiber size in the tibialis anterior (TA) muscle (as shown by morphometric analysis) (Fig. 1g, h), and a higher number of smaller fibers (Fig. 1i). Moreover, they had a 22.6% reduced grip strength of fore and hindlimb muscles (Fig. 1j), suggesting an attenuated function of muscles lacking *Dhx36*. Finally, we examined the regenerative capacity of *Dhx36* cKO muscles after injection of cardiotoxin (CTX) to induce muscle damage and regeneration. Immunohistological staining of eMyHC (a marker protein for newly formed, regenerating myofibers) on cross-sections of TA muscles harvested at 7 dpi revealed a significant reduction in the size of eMyHC+ fibers (by 57.9%) in cKO muscles (Fig. 1k–m), indicating defective muscle regeneration upon *Dhx36* inactivation. Overall, these results imply that *Dhx36* loss in the early developmental SC precursors and progeny affects both normal growth and regeneration of skeletal muscle in adults.

**Dhx36 inactivation in adult SCs impairs muscle regeneration.** To further test cell-autonomous functions of DHX36 in SCs during muscle regeneration, we crossed *Dhx36^{fl/fl}* mice with *Pax7^{CreER}/ROSA^{EYFP}* mice (Fig. 2a) to specifically induce *Dhx36* deletion in adult SCs. Genetic inactivation of *Dhx36* in SCs was induced by intraperitoneal (i.p.) injection of tamoxifen (Tmx) in adult (8–12-week-old) mice for five consecutive days followed by 5 days of chase (Fig. 2b). Almost complete deletion of *Dhx36* was

achieved in activated SCs of the *Dhx36* inducible knockout (iKO) as compared to Ctrl mice, at both protein (Fig. 2c) and mRNA levels (Supplementary Fig. 1f).

We next examined whether loss of *Dhx36* in SCs impacts adult muscle regeneration, by intramuscular injection of TA muscles with BaCl$_2$, which induced acute damage followed by regeneration to an extent similar to CTX injection (Fig. 2d)[31]. Tissue degeneration with abundant immune cell infiltrates was observed at 2.5 dpi in both Ctrl and iKO muscles. At 5 dpi, regenerating fibers with centrally localized nuclei were readily observed in Ctrl damaged muscles while they were rare in iKO damaged muscles, which still presented abundant immune cell infiltrates. At 7 dpi, regeneration in iKO muscles was still significantly compromised, with newly formed myofibers much smaller than those in Ctrl mice (Fig. 2d). eMyHC staining confirmed delayed muscle regeneration in iKO mice: at 5 dpi, both Ctrl and iKO injured muscles presented eMyHC+ fibers, which were larger in Ctrl mice, as shown by myofiber cross-sectional area (CSA) measurement; at 7 dpi, however, eMyHC+ fibers were no longer present in Ctrl but persisted in iKO muscles, consistent with the larger size of newly formed myofibers in Ctrl muscles (Fig. 2e, f and Supplementary Fig. 1g). At 28 dpi, regeneration of TA muscles had advanced in both Ctrl and iKO mice; however, this process was still severely compromised in iKO mice, which had significantly smaller new fibers as compared to Ctrl mouse muscles, consistent with the 22% reduced muscle weight in iKO mice (Fig. 2g). Thus, deletion of *Dhx36* in SCs markedly blunts adult muscle regeneration, indicating its requirement for this process.

**Dhx36 deletion principally impairs SC proliferative capacity.** To pinpoint the defects of SCs upon *Dhx36* deletion, we investigated SC behavior in vitro and in vivo. First, we found that the number of Pax7+ SCs in Ctrl and iKO adult mice did not differ significantly 4 weeks after Tmx administration (Fig. 3a), indicating that *Dhx36* deletion did not impact the homeostatic SC quiescent state. Consistently, Pax7 staining also revealed no difference in SC number on single myofiber explants from iKO or Ctrl mice (Supplementary Fig. 2a). To assess SC proliferative capacity, we measured EdU incorporation in SCs cultured for 2 days. While 66.7% of SCs were EdU+ in Ctrl mice, only 39.7% were positive in iKO mice (Fig. 3b), suggesting that the proliferative ability of SCs was compromised by *Dhx36* loss. These results were confirmed on single myofiber explants: at 2 days after isolation, iKO myofibers had 57.3% fewer Pax7+ SCs, and 65.0% fewer YFP+ SCs, as compared to Ctrl myofibers (Fig. 3c and Supplementary Fig. 2b). Moreover, in response to muscle damage, the number of Pax7+ SCs was also significantly decreased (by 52.9% decrease) in iKO vs Ctrl mice at 3 dpi (Fig. 3d). Finally, EdU incorporation in SCs

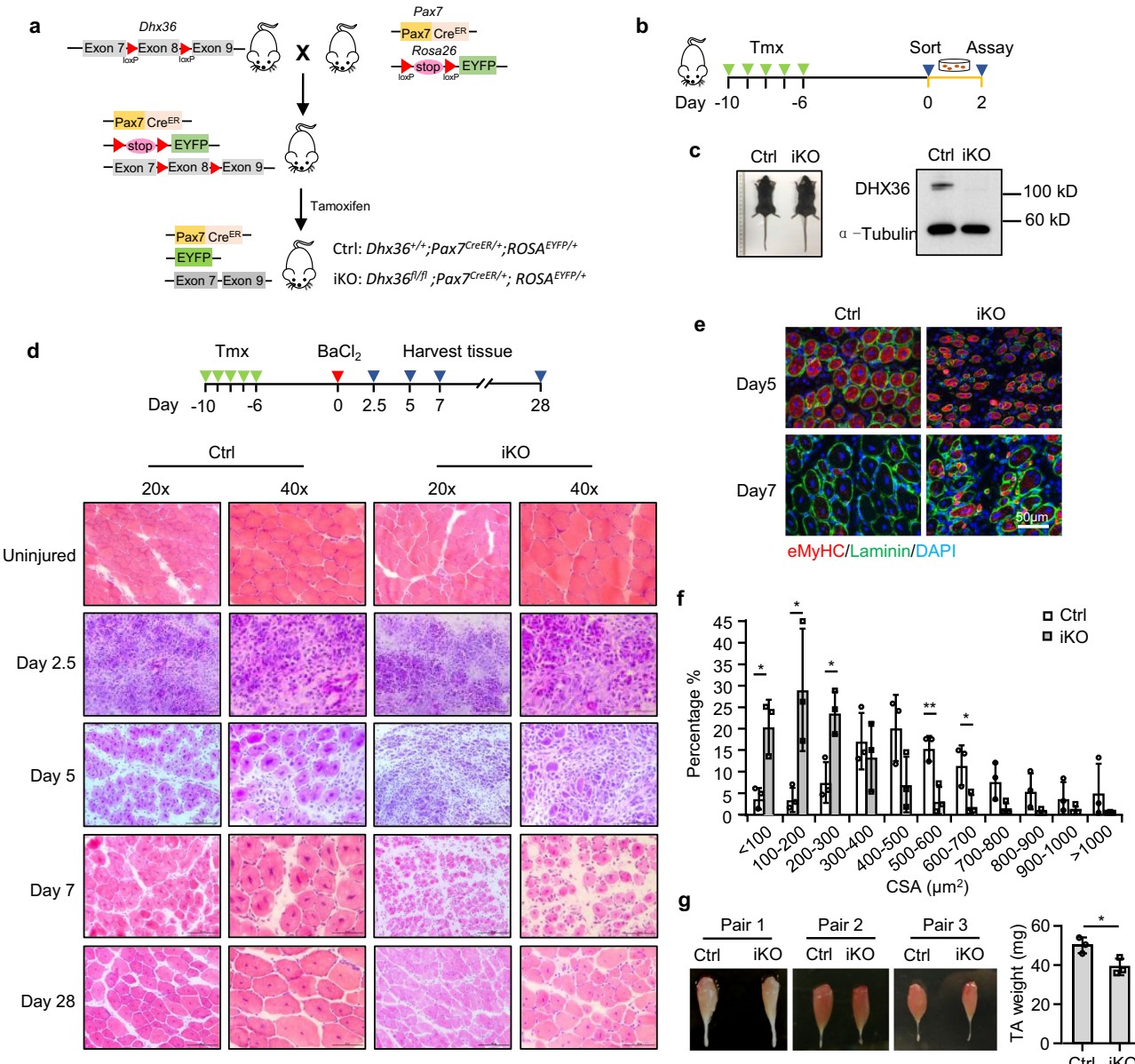

**Fig. 2 Dhx36 inactivation in adult SCs impairs muscle regeneration. a** Schematic illustration of the strategy to inactivate *Dhx36* in inducible knockout (iKO) mice. *Dhx36fl/fl* mice were mated with *Pax7CreER; ROSAEYFP* mice; the exon 8 of *Dhx36* was deleted in the iKO mice after Tamoxifen (Tmx) injection which also resulted in the removal of the stop signal for YFP at the Rosa26 site to allow the expression of YFP in iKO SCs. **b** Schematic illustration of five doses of Tmx injection to delete *Dhx36* in the iKO mouse. Freshly isolated SCs were collected at day 6 post-injection and cultured for 2 days. **c** Left: no obvious morphological difference was observed in representative Ctrl vs iKO mice. Right: loss of DHX36 protein was confirmed by western blot with α-Tubulin as the loading control. **d** Upper panel: schematic illustration of the injury-induced muscle regeneration scheme. BaCl₂ was injected into TA muscles of the above Ctrl or iKO mice 6 days post-Tmx injection to induce acute injury. The injected TA muscles were harvested at the designated times for the assessment of the regeneration process. Lower panel: H&E staining of the TA muscles collected at 2.5, 5, 7, and 28 days post injury. Scale bar = 100 μm (×20) or 50 μm (×40). **e** IF staining of eMyHC (red) and laminin (green) was performed on the TA muscles collected at 5 and 7 days post BaCl₂ injury. Nuclei were visualized by DAPI staining (blue). Scale bar = 50 μm. **f** CSAs of newly formed fibers were quantified from the above-stained sections and the distribution is shown, Ctrl in white bars and iKO in gray bars; *n* = 3 mice per group. <100 μm, *P* = 0.013; 100–200 μm, *P* = 0.038; 200–300 μm, *P* = 0.016; 500–600 μm, *P* = 0.009; 600–700 μm, *P* = 0.036. **g** Left: Representative images of TA muscles collected at 28 days post injury are shown. Right: The muscle weight from three pairs of mice, *P* = 0.031. Data are represented as mean ± s.d., Student's *t* test (two-tailed unpaired) was used to calculate the statistical significance (**f**, **g**): *\*P* < 0.05, \*\**P* < 0.01. Source data are provided as a Source Data file.

in vivo at 2.5 dpi in muscles of Ctrl or iKO mice (all carrying the Rosa EYFP reporter) showed that 31.8% of YFP+ SCs were EdU+ in Ctrl mice, but only 18.3% were in iKO mice (Fig. 3g), confirming that loss of *Dhx36* in SCs compromised their proliferating capacity both in vitro and after muscle injury.

To further investigate whether the impaired proliferation of *Dhx36*-iKO SCs was caused by a defect in quiescence exit or in

the first cell cycle entry (i.e., the activation step), SCs isolated from Ctrl or iKO mice were cultured with EdU at early time points. After 24 h, 10.4% of SCs were EdU+ in Ctrl mice, but only 3.0% were in iKO mice (Fig. 3e); consistently, on the single myofibers cultured for 36 h, 79.8% of SCs were EdU+ in Ctrl mice, and only 38.7% were in iKO mice (Fig. 3f). Intriguingly, co-staining of MyoD and Pax7 on single myofibers at 24 h after

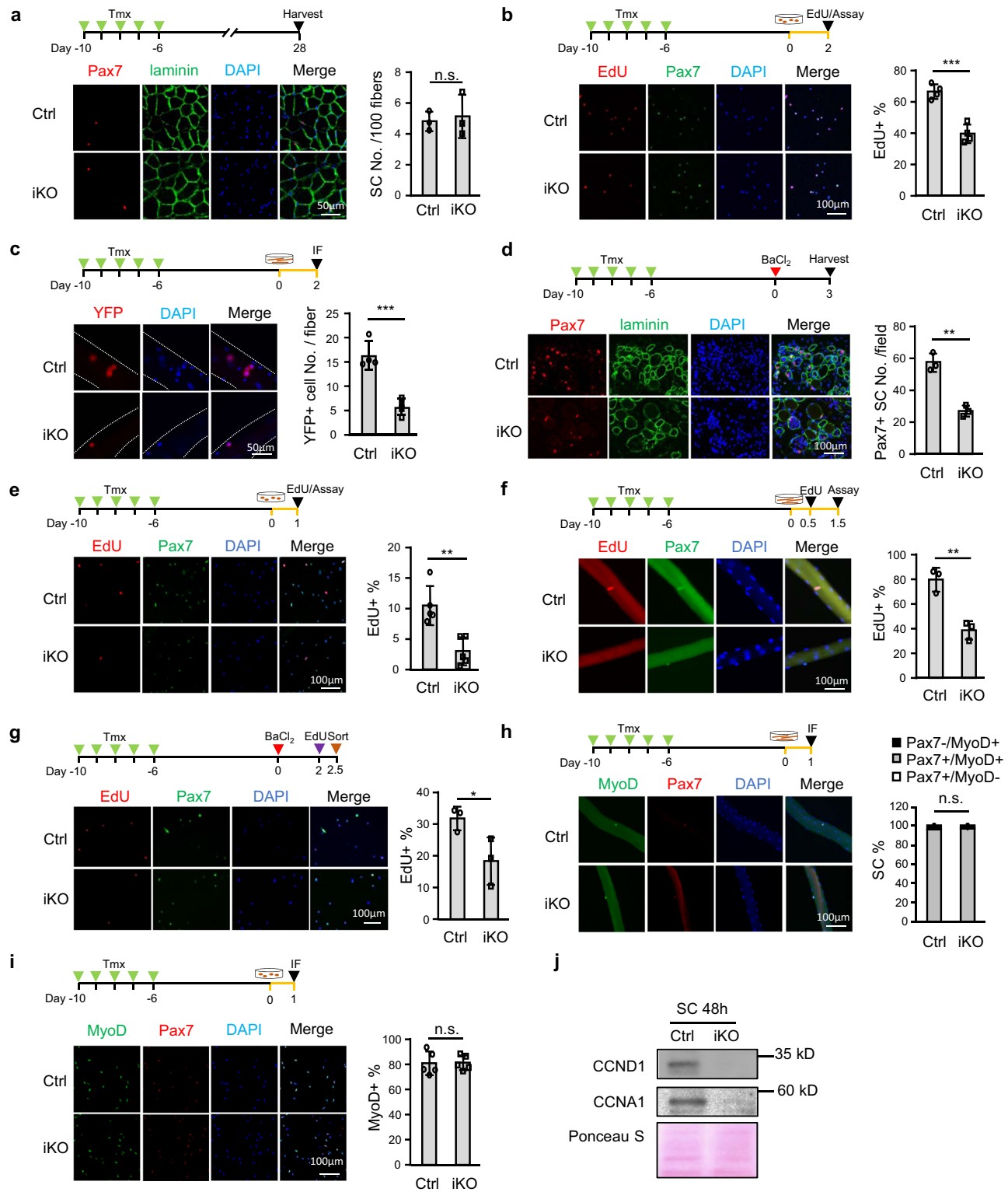

isolation revealed that nearly all SCs were MyoD+ in both Ctrl and iKO myofibers (Fig. 3h), suggesting that quiescence exit was not affected by *Dhx36* deletion. Further, after culturing freshly sorted SCs for 1 day, a similar percentage of Ctrl (81.0%) and iKO (81.4%) SCs were MyoD+ positive (Fig. 3i), and there was no significant difference of MyoD expression at either mRNA or protein levels between Ctrl and iKO SCs (Supplementary Fig. 2c, d). Moreover, these myofibers had a similar percentage of Ki67+ SCs (control, 94.4%; iKO, 93.4%), suggesting that the *Dhx36*-KO-

induced defect was not in cell cycle re-entry but rather in cell cycle progression (Supplementary Fig. 2e). To confirm this, we deleted *Dhx36* in C2C12 myoblasts by CRISPR-Cas9-mediated gene editing (Supplementary Fig. 2f). Indeed, homozygous deletion of *Dhx36* reduced EdU incorporation by 46% as compared to Ctrl clones (Supplementary Fig. 2g). Further cell cycle analysis by FACS revealed a severe G1 block in the KO cells after nocodazole treatment (Supplementary Fig. 2h); consistently, the cycling genes CCND1 and CCNA1 (which regulate the G1/S phase transition)

**Fig. 3 *Dhx36* deletion principally impairs SC proliferative capacity. a** IF staining of Pax7 (red) and laminin (green) on Ctrl and iKO TA muscles 4 weeks after Tmx injection. Scale bar = 50 μm; $n = 3$ mice per group. **b** SCs from Ctrl or iKO mice were cultured for 48 h. EdU was added to the culture medium 4 h before staining for EdU and Pax7. Scale bar = 100 μm; $n = 4$ mice per group. $P = 0.00038$. **c** IF staining of YFP on single myofibers isolated from Ctrl or iKO mice and cultured for 48 h. Scale bar = 50 μm; $n = 5$ mice per group. $P = 0.00073$. **d** IF staining of Pax7 (red) and laminin (green) on TA muscles at 3 dpi. Scale bar = 100 μm; $n = 3$ mice per group. $P = 0.0014$. **e** SCs from Ctrl and iKO mice were cultured for 24 h with EdU before EdU (red) and Pax7 (green) immunostaining. Scale bar = 100 μm; $n = 5$ mice per group. $P = 0.0027$. **f** Isolated myofibers were cultured with EdU for 36 h before immunostaining of EdU (red) and Pax7 (green). Scale bar = 100 μm; $n = 3$ mice per group. $P = 0.0045$. **g** Lower hindlimb muscles were injured by $BaCl_2$ followed by intraperitoneal injection of EdU at 48 h post injury. Freshly isolated SCs from lower hindlimb muscles were collected 12 h later and stained for EdU (red). Scale bar = 100 μm; $n = 3$ mice per group. $P = 0.0498$. **h** IF staining of Pax7 (red) and MyoD (green) was performed on myofibers cultured for 24 h. Scale bar = 100 μm; $n = 3$ mice per group. **i** IF staining of Pax7 (red) and MyoD (green) on SCs cultured for 24 h. Scale bar = 100 μm; $n = 5$ mice per group. **j** CCND1 and CCNA1 proteins were measured by western blot in Ctrl and iKO SCs cultured for 48 h. Positively stained cell numbers were quantified from ten randomly selected fields (**b**, **d**, **e**, **g**, **i**) or >15 fibers per mouse (**a**, **c**, **f**, **h**). Data are represented as mean ± s.d., Student's *t* test (two-tailed unpaired) was used to calculate the statistical significance (**a–i**): n.s., not significant, \*$P < 0.05$, \*\*$P < 0.01$, \*\*\*$P < 0.001$. Source data are provided as a Source Data file.

were also downregulated in iKO vs Ctrl SCs (Fig. 3j). Altogether, these findings demonstrate that *Dhx36* is indispensable for cell proliferation and cell cycle progression of SC-derived myoblasts during injury-induced muscle regeneration.

**Binding profiling of DHX36 uncovers its diversified roles**. The cytoplasmic localization of DHX36 in myoblasts (Fig. 1e) led us to speculate that DHX36 exerts its function as a post-transcriptional or translational regulator, consistent with its well-known rG4- and ARE-binding abilities and regulatory functions. Therefore, we analyzed C2C12 myoblasts by CLIP-seq to map the RNA interactome of endogenous DHX36 (see schematics of workflow, Supplementary Fig. 3a)[32]. Using an antibody against DHX36, we precipitated endogenous DHX36, at the expected size of ~115 kD (Fig. 4a, left), with the co-precipitated RNA-protein (RNP) complexes appearing 15–20 kD larger (Fig. 4a, right). The bound RNA fragments were recovered from the RNPs and transformed into small RNA libraries for next-generation sequencing. Through an in-house pipeline modified from previous publication[32], we identified a total of 5305 and 4982 binding sites from two biological replicates, which showed good reproducibility (Fig. 4b). Overall, 3662 sites (corresponding to 1262 individual genes) were shared between the two replicates (Fig. 4c and Supplementary Dataset 1). The majority of the binding peaks (3032; 82.7%) were mapped to protein-coding mRNAs, with a small portion of binding to non-coding RNAs (ncRNAs) (422), small nucleolar RNAs (snoRNAs) (142), or other RNA species (66) (Fig. 4d). Thus, DHX36 appears to have a predominant function in regulating mRNA processing. Close examination of mRNA binding revealed a large number of binding sites resided in coding sequences (CDS) (2040) and introns (1345) (Fig. 4e and Supplementary Dataset 1); however, the 5′ UTRs possessed the highest binding density (binding site number per 1000 nt), followed by CDS, 3′ UTRs and introns (Fig. 4f), suggesting that DHX36 carries out its dominant functions through binding and regulating 5′ UTRs. Notably, 216 transcripts had DHX36 binding sites in multiple regions (in their 5′ UTRs, 3′ UTRs, and CDS) (Fig. 4g, h and Supplementary Dataset 1). Gene ontology (GO) analysis of the 1262 genes associated with DHX36 binding, showed enrichment in "regulation of gene expression, epigenetic", "gene silencing", and "DNA conformation change" (Fig. 4i), suggesting that DHX36 had diversified gene-regulatory functions.

Further scanning of the binding motifs, based on all binding peaks by MEME[33], showed a strong enrichment of G-rich sequences; these were also the top-ranked motifs when the scanning was performed in the 5′ UTR, 3′ UTR, or CDS-binding sites alone (Fig. 4j and Supplementary Dataset 1). We next predicted whether these G-rich sequences were capable of forming rG4 structures using in-house python scripts (see

"Methods"). Among the top four DHX36-binding-motif-enriched regions within 5′ UTRs, 57% of Motif 1 were predicted to form an rG4 structure (Fig. 4k), highlighting that the primary function of DHX36 in 5′ UTRs is indeed related to rG4 binding. Intriguingly, unlike previous reports that show DHX36 binding to the canonical $G_3L_{1-7}$ (($G_3N_{1-7}$)$_3G_3$, N = A, U, C, or G) G4 sequences in 5′ UTRs[34], bulges and two quartets appeared to dominate in our predicted rG4s. We next experimentally validated the formation of rG4 on five selected targets with DHX36 binding on their 5′ UTRs (e.g., *Cd82, Hnrnpl, Mknk2, Nid2,* and *Pja2*) (Supplementary Dataset 4). Synthesized RNA oligonucleotides corresponding to the predicted rG4 formation sequences in each mRNA (Supplementary Fig. 4c) were used for ligand-enhanced fluorescence spectroscopy (Supplementary Fig. 4a, b). By probing with Thioflavin T (ThT), which becomes fluorescent in the presence of a G4 structure[35], the five RNA oligonucleotides exhibited an enhanced emission at the wavelength of 490 nm in K+ (which is known to stabilize G4 structure) as compared with Li+ (Supplementary Fig. 4d). Next, probing with another G4 ligand *N*-methylmesoporphyrin (NMM)[36,37] also showed that these targets could form rG4 with K+ (Supplementary Fig. 4e), as they exhibited an enhanced emission at 610 nm in K+ but not Li+ conditions. Moreover, in CD spectrum assay these five RNA oligonucleotides displayed a clearly enhanced negative peak at 240 nm and a positive peak at 262 nm in K+ vs Li+ conditions (Supplementary Fig. 4f), indicating the parallel topology of these rG4 structures[38]. Furthermore, the UV melting spectra showed the melting temperature (Tm) of the G4 structure (Supplementary Fig. 4g), confirming that rG4s could be formed in the DHX36-bound mRNAs. In contrast, in both 3′ UTR and CDS regions, the percentage of predicted rG4 structures was much lower (18% and 11.6%, respectively) (Fig. 4l-m). In addition to the G-rich sequences, one type of ARE (often defined as adjacent repeated AUUUA repeats)[39] was also found to be highly enriched (ranking no. 3) in the DHX36-binding regions in 3′ UTRs (Fig. 4l) but not in 5′ UTRs or CDS (Fig. 4j). Previously, DHX36 was shown to bind to ARE motifs and to regulate urokinase plasminogen activator (*uPA*) mRNA degradation[12]; our finding thus indicated that ARE binding may be a transcriptome-wide event that endows DHX36 with key regulatory functions. Indeed, a total of 292 AREs (within 154 genes) were bound by DHX36, with most in the 3′ UTRs (Supplementary Fig. 3b and Supplementary Dataset 1); GO analysis revealed that the genes with ARE–DHX36-binding sites in their 3′ UTRs were enriched for "mesenchyme development", "cell migration", and "cell morphogenesis involved in differentiation" processes, among others (Supplementary Fig. 3c and Supplementary Dataset 1). In addition to the above rG4 and ARE motifs, we noticed that C/U- rich motifs were highly ranked in both 5′ UTRs and

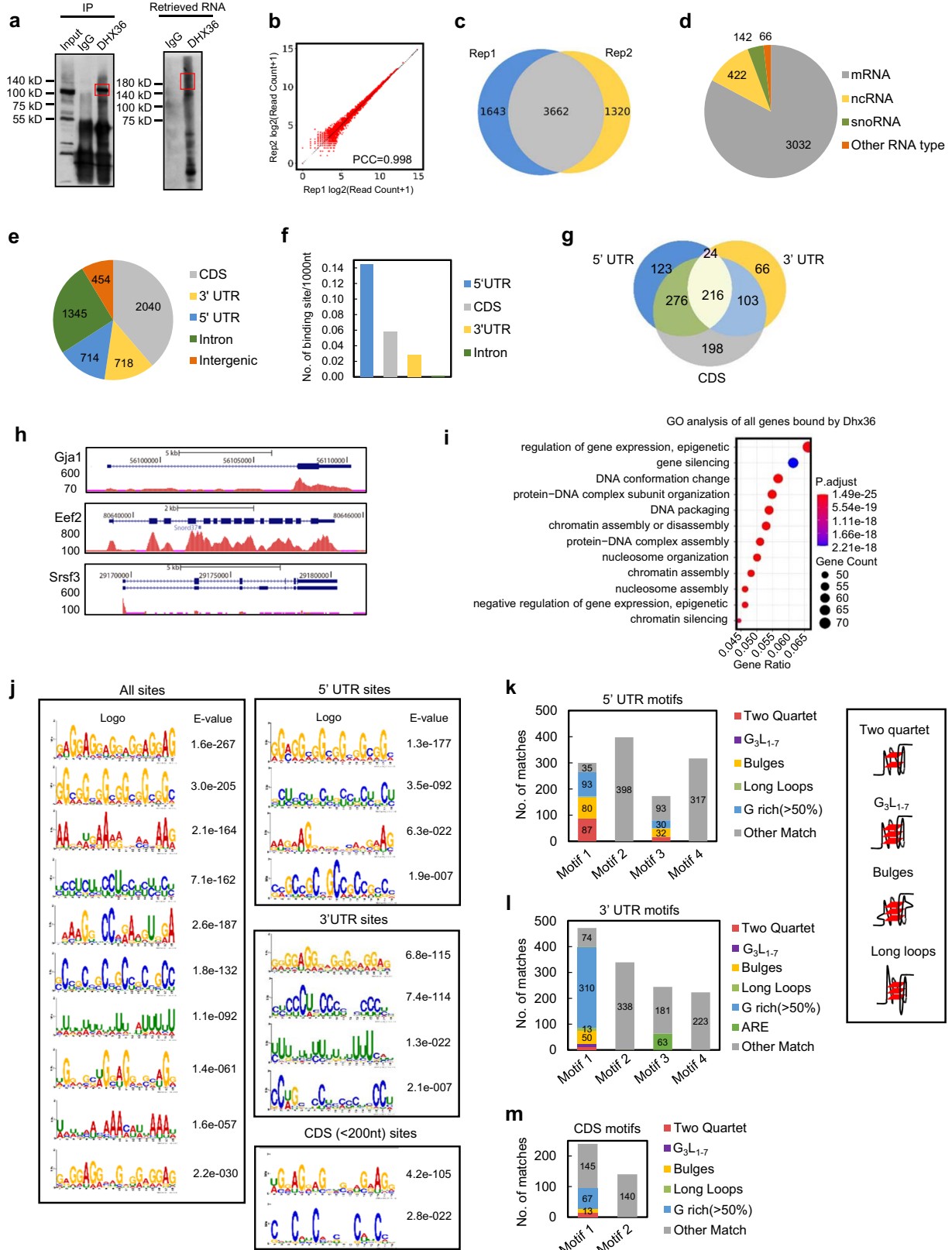

3′ UTRs, although no specific sequences could be identified (Fig. 4j). Nevertheless, these sequences were predicted (using RNAshapes[40]) to form a very similar secondary structure, defined by continuous hairpins with flanking stems (Supplementary Fig. 3d, e), suggesting that DHX36 recognizes other specific secondary structures besides G4 and ARE. Lastly, on the binding

regions of CDS, 8-nt binding motifs were also identified (Supplementary Fig. 3f); notably, these motifs were similar to the binding motif for splicing regulators, including SRSF1, SRSF2, TRA2A, TRA2B, and HNRNPA2B1 (Supplementary Fig. 3g), suggesting a possibility for DHX36 involvement in splicing regulation. Similarly, we also noticed that DHX36-binding sites in

**Fig. 4 Binding profiling of DHX36 uncovers its diversified roles. a** CLIP-seq was performed in C2C12 myoblasts and western blot showed an evident amount of DHX36 protein was immunoprecipitated with the antibody against DHX36 but not IgG. The above-retrieved RNAs were labeled with biotin for the visualization and size selection. The indicated RNAs were recovered and used for sequencing. **b** Read coverages in the two CLIP-seq biological replicates displayed a high correlation. PCC Pearson correlation coefficient. **c** A large number of identified binding sites (3662) were shared in the two biological replicates. **d** The categories of the DHX36-associated RNAs are shown. **e** The distribution of the overlapped sites on different mRNA regions. **f** The density of DHX36 binding sites in the designated regions of mRNAs. **g** Venn diagram showing the number of mRNAs bound by DHX36 in different mRNA regions. **h** Genome snapshots to illustrate DHX36 binding in 3′ UTR, CDS and 5′ UTR on three selected transcripts. **i** The GO analysis results for all genes with DHX36 binding ordered by gene ratio (proportion of genes annotated for each GO term). Dots are colored by adjusted *P* value (degree of enrichment) and their size corresponds to the gene counts annotated to each GO term. **j** MEME identifies the top enriched motifs in all binding sites or sites residing in 5′ UTRs, 3′ UTRs, or CDS. The *E*-values measuring the statistical significance of each motif identified by MEME are shown. **k** Potential rG4 formation was predicted in the top four enriched DHX36 binding motifs in 5′ UTRs; the number of each subtype of rG4s (illustrated on the right) is shown. **l** Potential rG4 formation or ARE was predicted in the top four enriched DHX36 binding motifs in 3′ UTRs; the number of each subtype of rG4s or AREs is shown. **m** Potential rG4 formation was predicted in the top two enriched DHX36 binding motifs within CDS; the number of each subtype of rG4s is shown.

introns (Supplementary Fig. 3h and Supplementary Dataset 1) were enriched for a motif resembling the C box motif (RUGAUGA) of C/D box snoRNAs (Supplementary Fig. 3i). Further, using snoSCAN[41], some of the binding sites overlapped either annotated (52/234) or predicted snoRNAs (21/182), indicating an intriguing connection between DHX36 and snoRNAs, which was previously reported for other DEAD or DEAH box RNA helicases[42]. Altogether, the above global profiling of DHX36 binding to mRNA has provided a comprehensive view of endogenous DHX36 binding in myoblasts, demonstrating the predominant binding of DHX36 to rG4 structures in 5′ UTRs of transcripts, while revealing previously unknown aspects of DHX36 involvement in mRNA metabolism.

**DHX36 facilitates translation by binding to 5′ UTR rG4s.** The enrichment of DHX36 binding at rG4 sites of 5′ UTRs suggested its potential involvement in global translational regulation by unfolding the 5′ UTR rG4 in myoblasts. Accordingly, we performed polysome profiling coupled with RNA-seq to examine the translational status of mRNAs in C2C12 myoblasts with *Dhx36* deletion (*Dhx36* KO) (Fig. 5a). Briefly, *Dhx36* KO or Ctrl cell lysates were sedimented by sucrose density centrifugation and fractionated into the supernatant, small subunits (40S), large subunits (60S), monosomes (80S), and polysomes (associated to ≥5 ribosomes thus considered to undergo active translation) fractions (Fig. 5a). The global absorbance profiles of ribosomes showed no obvious difference between the Ctrl and KO cells (Fig. 5b), suggesting that DHX36 does not affect global protein synthesis; this is consistent with a prior report in HEK293T cells in which *Dhx36* knockdown did not have a major impact on polysome absorbance profiles[43].

To identify mRNAs whose translational efficiency (TE) might be affected by *Dhx36* loss, total mRNAs from the cytoplasmic fraction and the polysome-associated mRNA fractions were analyzed by high-throughput sequencing. TE for each mRNA was calculated by normalizing the polysome-associated mRNA to the total cytoplasmic mRNA level (see "Methods"). The log2 values of TE fold change in KO vs control (log2(ΔTE)) were highly reproducible in the two biological replicates (Supplementary Fig. 5a and Supplementary Dataset 2; *n* = 19179), with a large proportion of the mRNAs exhibiting a similar TE change trend (e.g., either up- or downregulated in both replicates) (Supplementary Fig. 5a, red dots, *n* = 15,068), with a Spearman correlation coefficient (SCC) of 0.956. Of these 15,068 mRNAs, *Dhx36* loss resulted in upregulated TE values (fold change ≥ 1.5; TE-up) in 1697, and downregulated TE values (fold change ≤0.66; TE-down) in 1796 (Fig. 5c and Supplementary Dataset 2). GO analysis revealed that the TE-up genes were enriched in processes such as "protein–DNA complex assembly" or "chromatin

assembly", and the TE-down genes were enriched in "pattern specification" or "positive regulation of neurogenesis", among others (Supplementary Fig. 5b, c). To further elucidate the TE changes caused by direct DHX36 binding, the above data were intercepted with DHX36 binding targets from the CLIP-seq, which uncovered a much higher number (76 vs 38) of TE-down vs TE-up transcripts with DHX36 binding (Fig. 5d and Supplementary Dataset 2). These results suggested that DHX36 selectively promoted the translation of a portion of target mRNAs. GO analysis showed that this subset of TE-down genes was enriched in "skeletal system development" and "aging", while the TE-up genes were enriched in "nucleosome assembly" and "chromatin assembly", among other processes (Fig. 5e, f). Supporting the hypothesis that this translational effect is largely related to 5′ UTR binding, we found that DHX36 binding to 5′ UTRs conferred a stronger effect on TE change (a total of 73 transcripts with TE changes; Fig. 5g) as compared to its binding within 3′ UTRs (19) and CDS (46) (Fig. 5h, i). Furthermore, among these 73 transcripts, 61 were TE-down, and 12 were TE-up, upon *Dhx36* loss (Supplementary Dataset 2), suggesting that DHX36 binding to the 5′ UTR principally promotes translational efficiency. On the other hand, DHX36 binding to the 3′ UTR may also increase the TE: we observed 1 TE-up vs 18 TE-down mRNAs upon *Dhx36* loss (Fig. 5h and Supplementary Dataset 2). Interestingly, DHX36 binding within CDS regions appeared to confer both increased and decreased TE, as about an equal number (22 vs 24) of transcripts were up-or downregulated upon *Dhx36* loss (Fig. 5i and Supplementary Dataset 2). In line with the above findings, analyses with kernel density estimate showed slightly lower log2(ΔTE) values were associated with 5′ UTR-bound mRNAs as compared to values of 3′ UTR- or CDS-bound mRNAs (Fig. 5j).

As a further test of whether the TE-promoting effect by DHX36 is conferred by 5′ UTR rG4 structures, we found indeed a large number of TE-down mRNAs with 5′ UTR DHX36 binding contained rG4 structures (58/61); a closer look at the total of 65 binding sites on these mRNAs revealed that the majority (37/65) possessed bulges as the dominant rG4 structure (Fig. 5k). In contrast, only one TE-up mRNA with 5′ UTR DHX36 binding contained rG4 structure. When 3′ UTR-bound mRNAs were analyzed, some correlation was found between rG4 formation and the number of TE-down transcripts (14/18) (Fig. 5h), but this was weaker at CDS (14/24) (Fig. 5i). Furthermore, the rG4 structures appeared again to be dominated by bulges and two quartets (Fig. 5l, m and Supplementary Dataset 2).

In addition, the TE changes in 3′ UTR-bound genes had no clear correlation with the presence of ARE (only 5/19 mRNAs contained AREs) (Fig. 5n). Furthermore, we found much lower than background rG4 folding energy, $\Delta G^0_{rG4}$ (i.e., more stably folded rG4 structures) of mRNAs with 5′ UTR binding in the TE-

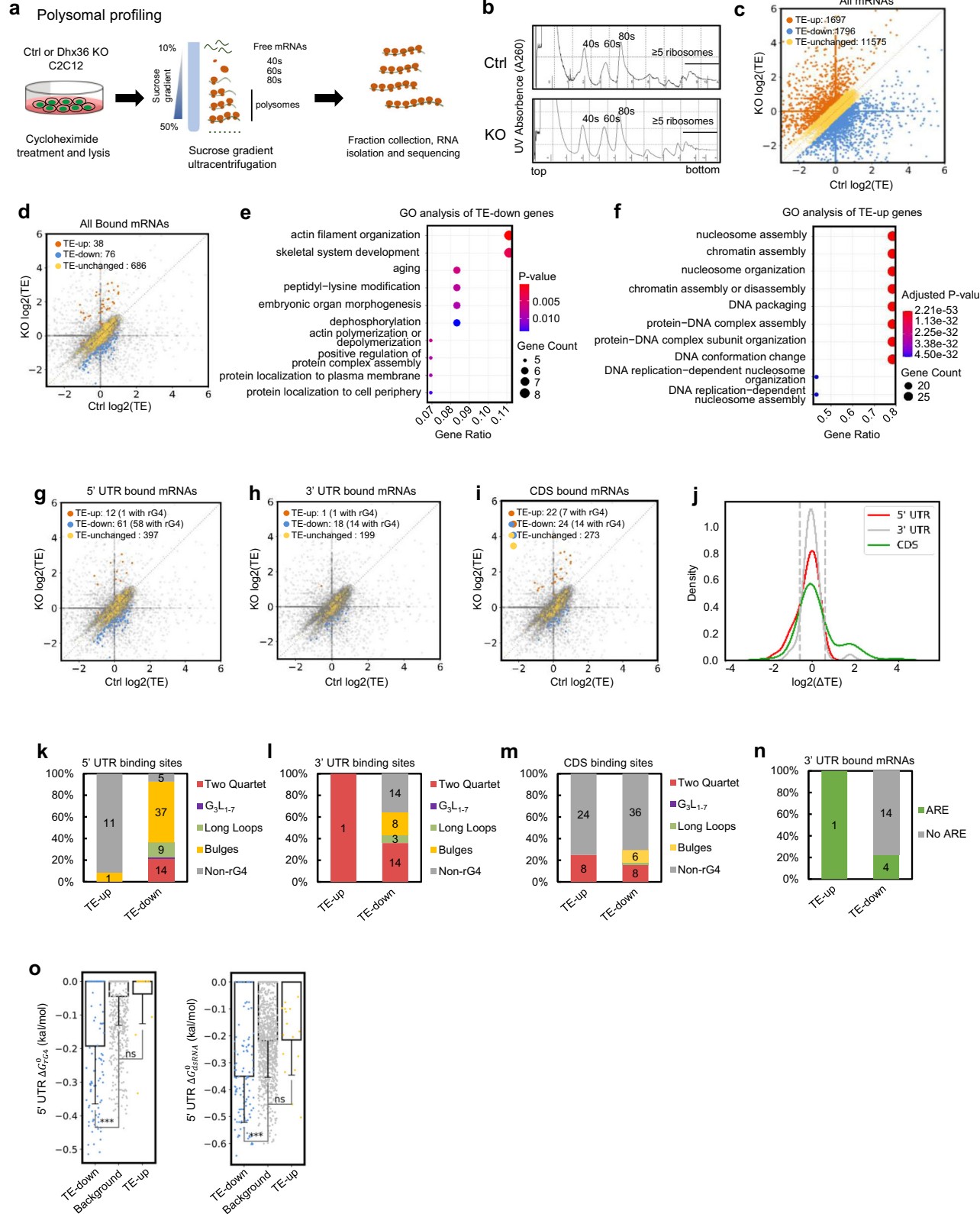

down subset than those in the TE-up subset. Finally, the sequence context favored the formation of rG4 structures over dsRNAs as $\triangle G^0_{rG4}$ displayed a much larger fold difference compared to background (mean energy value 4.3 fold, $P = 2.64 \times 10^{-23}$) than $\triangle G^0_{dsRNA}$ (mean energy value 1.61 fold, $P = 1.365 \times 10^{-16}$) (Fig. 5o). Taken together, these findings revealed that rG4 structures

in 5′ UTRs are a key determinant for the promoting function of DHX36 in mRNA translation.

**Dhx36 loss exerts a potential effect on total mRNA abundance.** To investigate whether DHX36 could modulate mRNA abundance, we next explored the global effect of *Dhx36* loss on total

**Fig. 5 DHX36 facilitates translation by binding to the 5′ UTR rG4s. a** Workflow for polysome profiling performed in Ctrl or *Dhx36*-KO C2C12 myoblast cells. **b** The polysome profiles in Ctrl or KO cells reveal no global changes of translation by *Dhx36* loss. In all, 10–50% sucrose gradient was applied to achieve separation of polysomes. The peaks for small (40S) and large (60S) subunits, as well as monosome (80S), ≥5 ribosomes are indicated. **c** The translational efficiency (TE) in Ctrl or KO cells was calculated and the scatterplot of log2(TE) values of all mRNAs ($n = 15,068$ mRNAs with consistent changes in two biological replicates) is shown. The numbers of mRNAs with up- (≥1.5-fold in KO vs Ctrl, orange dots) or downregulated (≤0.66-fold in KO vs Ctrl, blue dots) or unchanged (yellow) TE are shown. **d** The above analysis was performed on all DHX36-bound mRNAs. **e, f** GO analysis result for TE-up or -down genes in panel **d**. **g–i** Scatterplot showing log2(TE) in KO vs Ctrl in 5′ UTR, 3′ UTR, or CDS-bound transcripts. **j** Kernel density estimates on log2 (ΔTE) values of mRNAs with DHX36 binding only in 5′ UTR, 3′ UTR, or CDS. ΔTE: the TE alteration in KO vs Ctrl. **k–m** Prediction of rG4 formation in binding sites on 5′ UTR, 3′ UTR, and CDS-bound mRNAs was conducted and the number of binding sites possessing each subtype is shown in TE-up or TE-down mRNAs. **n** Prediction of ARE sites in binding sites on 3′ UTR-bound mRNAs was conducted and the number of mRNAs possessing ARE is shown. **o** 5′ UTR region predicted folding energies of rG4 and dsRNA structures in TE-up ($n = 111$ mRNAs) and TE-down ($n = 16$ mRNAs) genes with 5′ UTR binding compared with background genes ($n = 1075$ mRNAs) which have neither TE change nor DHX36 binding. One-tailed Mann–Whitney test was used to calculate the statistical significance between folding energy values: n.s., not significant ($P \geq 0.05$), ***$P < 0.001$. rG4 TE down vs background, $P = 2.64 \times 10^{-23}$; dsRNA TE-down vs background, $P = 1.365 \times 10^{-16}$. Data are represented as mean values ± s.d.

mRNA abundance using the 15,068 genes with a similar TE change trend (Supplementary Fig. 5a, red dots and Supplementary Dataset 3). Only a small number of mRNAs were upregulated (125; fold change (FC) ≥1.5) or downregulated (121, FC ≤0.66) in *Dhx36* KO vs Ctrl cells (Fig. 6a and Supplementary Dataset 3). The mRNA-up genes were enriched for GO terms including "tube morphogenesis" and "regulation of cytoskeleton organization", while mRNA-down genes were associated with processes such as "nucleosome assembly" and "chromatin assembly" (Supplementary Fig. 6a, b). Interestingly, binding within the CDS or the 3′ UTRs conferred a stronger effect on mRNA abundance than binding in the 5′ UTRs, as 92/99 vs 63/99 mRNAs were altered, respectively (Fig. 6b–d and Supplementary Dataset 3). Specifically, 47 5′ UTR-bound mRNAs were upregulated, while 16 were downregulated, suggesting that 5′ UTR binding decreased mRNA abundance. However, the opposite trend was observed for mRNAs bound by DHX36 at the 3′ UTRs (37 upregulated vs 55 downregulated) or CDS (36 upregulated vs 63 downregulated), suggesting that DHX36 binding on 3′ UTRs or CDS may cause an increase in mRNA abundance. Similar conclusions can be drawn when further plotting the kernel density estimate of the log2(ΔFPKM-total) values of transcripts with only 5′ UTR, 3′ UTR, or CDS-binding sites (Fig. 6e). When further examining the correlation between rG4 formation and the abundance change of mRNAs, we found a particularly strong correlation between rG4 in 5′ UTR with mRNA increase upon *Dhx36* loss (43/47 upregulated transcripts possessed rG4s; Fig. 6b), implying that DHX36 binding to 5′ UTR rG4 may lead to decreased mRNA level. Although DHX36 binding to 3′ UTR and CDS appeared to increase mRNA levels, it is hard to conclude whether this was connected to rG4 binding, as only 27/55 and 29/63 transcripts contained a predicted rG4. Among all the rG4s residing in the described binding sites, bulges and two quartets, but not canonical rG4 motifs, dominated (Fig. 6f, g). Similar to Fig. 5o, when computing the 5′ UTR $\triangle G^{0}_{rG4}$ and $\triangle G^{0}_{dsRNA}$, we found $\triangle G^{0}_{rG4}$ was much lower than the background in the mRNA-up genes (mean energy value, 3.19 fold, $P = 4.09 \times 10^{-15}$, compared to 1.43 fold, $P = 4.11 \times 10^{-10}$), indicating that formation of rG4 structures is favored over formation of dsRNAs at the 5′ UTRs of these transcripts (Fig. 6i). We next investigated the potential connection between the presence of AREs at the 3′ UTR and mRNA abundance, as DHX36 binding to AREs affects RNA stability[12]. Unexpectedly, no strong correlation was found between the presence of ARE and mRNA abundance in 23 of the 92 3′ UTR-binding transcripts with AREs in the binding sites (Fig. 6j). However, by further computing the $\triangle G^{0}_{dsRNA}$ of 3′ UTR AREs from Fig. 6j, AREs in both up- and downregualted mRNAs exhibited significantly lower folding energy value than background (Fig. 6k);

the ambiguity of 3′ UTR AREs in regulating mRNA abundance, therefore, deserves further investigation.

Lastly, to exclude the effect of mRNA changes on TE decrease upon *Dhx36* loss, we compared the divergence at the transcriptional and translation levels. Among 114 DHX36-bound genes with significant TE changes, 5 out of 38 TE-up genes (blue dots) and 16 out of 76 TE-down genes (yellow dots) showed constant mRNA levels in KO cells vs Ctrl cells (Fig. 6l, left). With a stringent threshold of TE fold change which was <0.5, only five mRNAs, *Ubq1n1, Nfix, Gnai2, Fzd2*, and *Zdhhc16* showed significantly decreased TE upon *Dhx36* loss (Fig. 6m), suggesting that they might be canonical translational regulatory targets of DHX36 in myoblasts that merited further investigation.

**DHX36 binds to *Gnai2* mRNA via the 5′ UTR rG4.** After our system-wide analyses uncovered DHX36 as a major translation regulator through binding to 5′ UTR rG4, we next aimed to elucidate the downstream effectors of its pro-proliferative function in myoblasts by selecting a target transcript as a proof-of-principle (Fig. 6l, right). Several identified DHX36 targets were related to cell proliferation regulation, such as *Yap1, Gnb1, Adam10*, and *Gnai2* (Supplementary Fig. 5d, e). We chose *Gnai2* mRNA as it showed clear DHX36 binding on the rG4 sites within its 5′ UTR (Figs. 6m, 7a, b and Supplementary Fig. 7a). Of note, previous studies showed that constitutive activation of GNAI2 resulted in muscle hypertrophy and accelerated injury-induced muscle regeneration while knocking-out *Gnai2* reduced muscle size and impaired muscle regeneration[44,45]; to some degree, this phenocopies the impaired muscle regeneration in *Dhx36* mutant mice, suggesting that *Gnai2* could be a target of DHX36. Moreover, protein level of GNAI2 was increased upon SC activation and maintained during cell proliferation (Fig. 7c), which was concomitant with the dynamic expression of the DHX36 protein (Fig. 1c).

To validate the rG4 formation in the identified site of *Gnai2* 5′ UTR, we found that the G-rich region was very conserved among multiple species including mouse, human, chimp, rhesus monkey, and cow (Supplementary Fig. 8a) and that it was predicted to form a canonical rG4 structure. ThT probing showed that the 5′ UTR of WT *Gnai2* exhibited an enhanced emission at 490-nm wavelength in K+ vs Li+ conditions; notably, if all GGGs in the site were mutated to GAGs, the formation of G-quadruplexes was disrupted and the enhanced emission was largely lost in K+ condition (Fig. 7d and Supplementary Fig. 4a). Probing with the additional G4 ligand NMM[36,37] confirmed that the WT but not mutated 5′ UTR formed an rG4 in K+ condition (Supplementary Figs. 4b and 8b). Lastly, targeted G4 formation of *Gnai2* 5′ UTR was also examined by reverse-transcriptase stalling

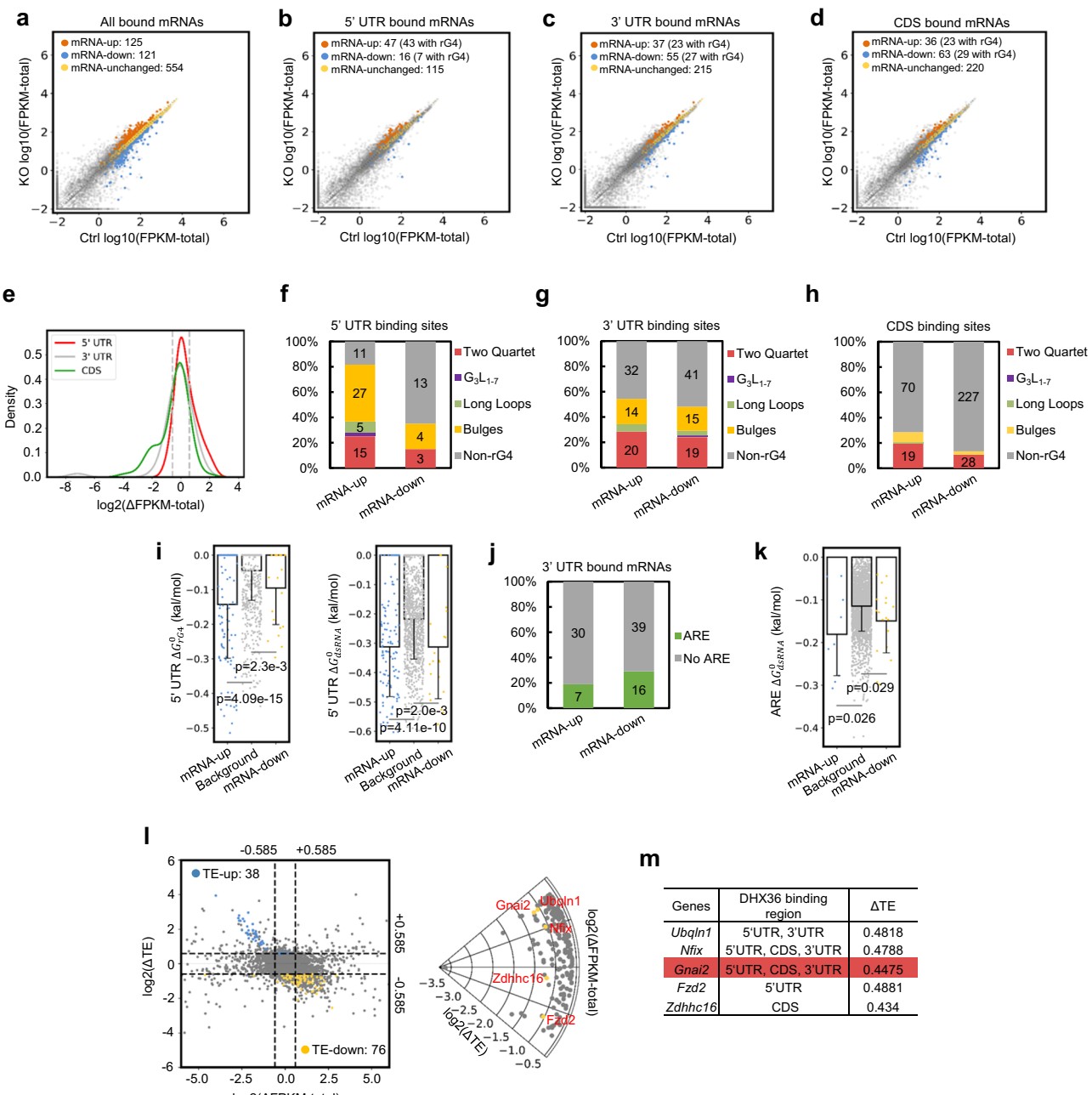

**Fig. 6 Dhx36 loss exerts a potential effect on total mRNA abundance. a–d** Scatterplot showing the total mRNA abundance (log10(FPKM-total)) values of all, 5′ UTR, 3′ UTR or CDS DHX36-bound mRNAs in Ctrl vs KO; the number of up- (KO vs Ctrl ≥1.5 fold), down-(≤0.66 fold) or unchanged mRNAs are shown. **e** Kernel density estimates showing log2(ΔFPKM-total) values of mRNAs with DHX36 binding only in 5′ UTRs, 3′ UTRs, or CDS. ΔFPKM-total: the mRNA level alteration in KO vs Ctrl. **f–h** Prediction of rG4 formation in the binding sites of 5′ UTR, 3′ UTR, and CDS-bound mRNAs was conducted and the number of sites possessing each subtype is shown in up- or downregulated mRNAs. **i** 5′ UTR region predicted folding energies of rG4 and dsRNA structures in up and downregulated mRNAs with 5′ UTR binding, $n = 116$ (upregulated mRNAs), 1075 (background mRNAs), 22 (downregulated mRNAs). Data represent mean values ± s.d. **j** The number of 3′ UTR-bound mRNAs possessing ARE sites is shown. **k** Predicted folding energies of dsRNAs in up- or downregulated mRNAs with 3′ UTR ARE binding, $n = 9$ (upregulated mRNAs), 2499 (background mRNAs), 18 (downregulated mRNAs). Data represent mean values ± s.d. **l** Left: Scatterplot illustrating TE (y axis) and mRNA (x axis) alterations in KO vs Ctrl with a threshold of ±0.585 (dashed line) for 7300 genes with FPKM-total values larger than 1 in either Ctrl or KO condition. Blue dots: TE-up genes with DHX36 binding; yellow dots: TE-down genes with DHX36 binding. Gray dots: the genes with no significant TE change or no DHX36 binding. Right: A total of five bound genes (yellow dots) including Ubq1n1, Nfix, Gnai2, Fzd2, and Zdhhc16 (red font) showed constant mRNA and decreased TE levels in KO vs Ctrl using a more stringent threshold (log2(ΔTE) <−1, ΔTE<0.5). Gray dots: 197 genes with significant TE decrease and constant mRNA levels but with no DHX36 binding. **m** DHX36 binding regions and ΔTE values are shown for the five genes in panel **l** and Gnai2 is highlighted. One-tailed Mann–Whitney test was used to calculate the statistical significance between folding energy values in panels **i** and **k** with P values shown in the figure.

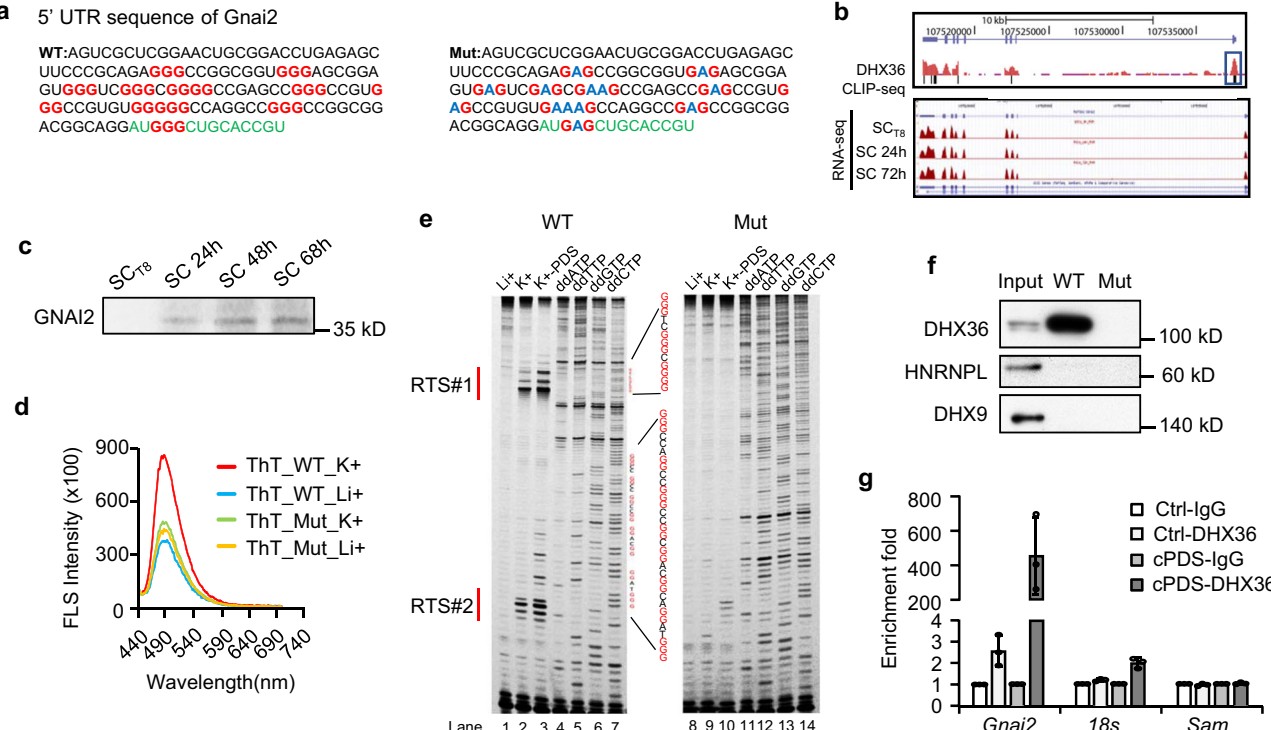

**Fig. 7 DHX36 binds to *Gnai2* mRNA via the 5′ UTR rG4. a** Sequence of the WT or mutant (Mut) *Gnai2* mRNA 5′ UTR. Green color denotes the coding sequence. All GGGs are highlighted in red. Mutated Gs are highlighted in blue. **b** Top: Genomic snapshot of DHX36 CLIP-seq track showing its binding at *Gnai2* mRNA. Bottom: Snapshot of RNA-seq tracks showing *Gnai2* mRNA levels in freshly isolated SCs, SCs cultured for 24 or 72 h. **c** GNAI2 protein levels in SCs cultured for the indicated time points were examined by western blot. **d** WT or Mut *Gnai2* 5′ UTR RNAs were treated with 150 mM Li+ or K+ together with 1 μM ThT and excited at 425 nm. Plot of the intensity of Fluorescent spectrum (FLS) with the wavelength from 440 to 700 nm. Red line: WT RNA treated with K+; blue line: WT RNA treated with Li+; green line: Mut RNA treated with K+; orange line: Mut RNA treated with Li+. **e** WT or Mut *Gnai2* 5′ UTR RNAs were treated with 150 mM Li+, 150 mM K+, or 150 mM K+ plus 2 μM PDS. Reverse transcriptase was stalled at the rG4 formation site to form reverse stalling site (RTS). The cDNAs generated by the above reverse transcription were run on a Dideoxy sequencing gel. Left: In WT *Gnai2* 5′ UTR RNA, two major RTS (highlighted as RTS #1 and #2) sites were identified. Right: no RTSs were detected in Mut *Gnai2* 5′ UTR RNA. **f** RNA pull-down was performed with in vitro transcribed and biotin-labeled WT or Mut *Gnai2* 5′ UTR RNAs and the retrieved DHX36 protein was detected by western blot. HNRNPL and DHX9 were used as negative controls. **g** C2C12 myoblast cells were treated with Ctrl (same volume of DMSO in the culture medium as cPDS) or 5 μM cPDS for 24 h and RNA immunoprecipitation (RIP) was performed to detect retrieved *Gnai2* transcripts. *18s* and *Sam* transcripts were used as negative controls. Data represent the average of three independent experiments ± s.d. (**g**). Source data are provided as a Source Data file.

(RTS) gel analysis[46] which used WT or mutated RNAs as the templates for reverse transcription in conditions with Li+, K+, or K+ plus PDS (pyridostatin, a G4 ligand which can further stabilize G4 structure); this assay detects the stalling caused by the formation of G4 structures (Supplementary Fig. 8c). Two very strong stalling sites (RTS#1 and RTS#2) were detected for the WT 5′ UTR under G4 formation conditions but not for the mutated 5′ UTR (Fig. 7e). Altogether, these results confirm the formation of rG4 structures at the *Gnai2* 5′ UTR.

Next, to validate the interaction between the above-identified rG4 site and DHX36, we performed RNA pull-down assay and found that the WT 5′ UTR sequence retrieved a significant amount of DHX36 protein, which was completely lost with the GGG-to-GAG mutated sequence (Fig. 7f and Supplementary Fig. 8e). Consistently, RNA immunoprecipitation (RIP) experiment showed that the DHX36 antibody effectively retrieved *Gnai2* mRNA but not control RNAs including *18s and Sam*[31] (Fig. 7g and Supplementary Fig. 8d). Moreover, when myoblast cells were treated with carboxyPDS (cPDS, a derivative of PDS which stabilizes rG4)[47], the interaction between DHX36 and *Gnai2* mRNA was significantly enhanced (as detected by RIP; Fig. 7g). Altogether, these results strengthen the finding that DHX36 interacts with the 5′ UTR of *Gnai2* through its rG4 site.

**DHX36 promotes *Gnai2* translation by resolving the 5′ UTR rG4.** To further investigate the translational control exerted by DHX36 in myoblasts, we found that GNAI2 protein, but not *Gnai2* mRNA, was significantly reduced in activated SCs from *Dhx36*-iKO vs Ctrl mice (Fig. 8a), as well as in proliferating *Dhx36*-KO C2C12 vs Ctrl myoblasts (Fig. 8b). Further, treatment of the *Dhx36*-KO cells with the proteasome inhibitor MG132 did not increase the GNAI2 protein level (Fig. 8c), suggesting that GNAI2 decrease in the KO cells was not caused by protein degradation. Moreover, we found that GNAI2 protein expression was significantly downregulated in SCs or C2C12 myoblasts with cPDS treatment while the mRNA level was increased or unaltered (Fig. 8d and Supplementary Fig. 8f), suggesting that GNAI2 protein level was indeed regulated by the G4 structure. Interestingly, we also found that SC proliferation was significantly decreased after cPDS treatment (by 26% with 2.5 μM cPDS, and by 54% with 5 μM cPDS; Supplementary Fig. 8g). To further examine the in cellulo rG4 formation, we stained SCs with an rG4 specific fluorescent ligand QUMA-1[48]. Interestingly, despite increasing DHX36 levels, we did not observe a corresponding decrease of rG4 signals in ASC 24 h vs SC-T8 cells (Supplementary Fig. 9a, b), probably indicating that DHX36 is not the sole rG4 regulator in SC cell cycle re-entry. Nevertheless, we observed

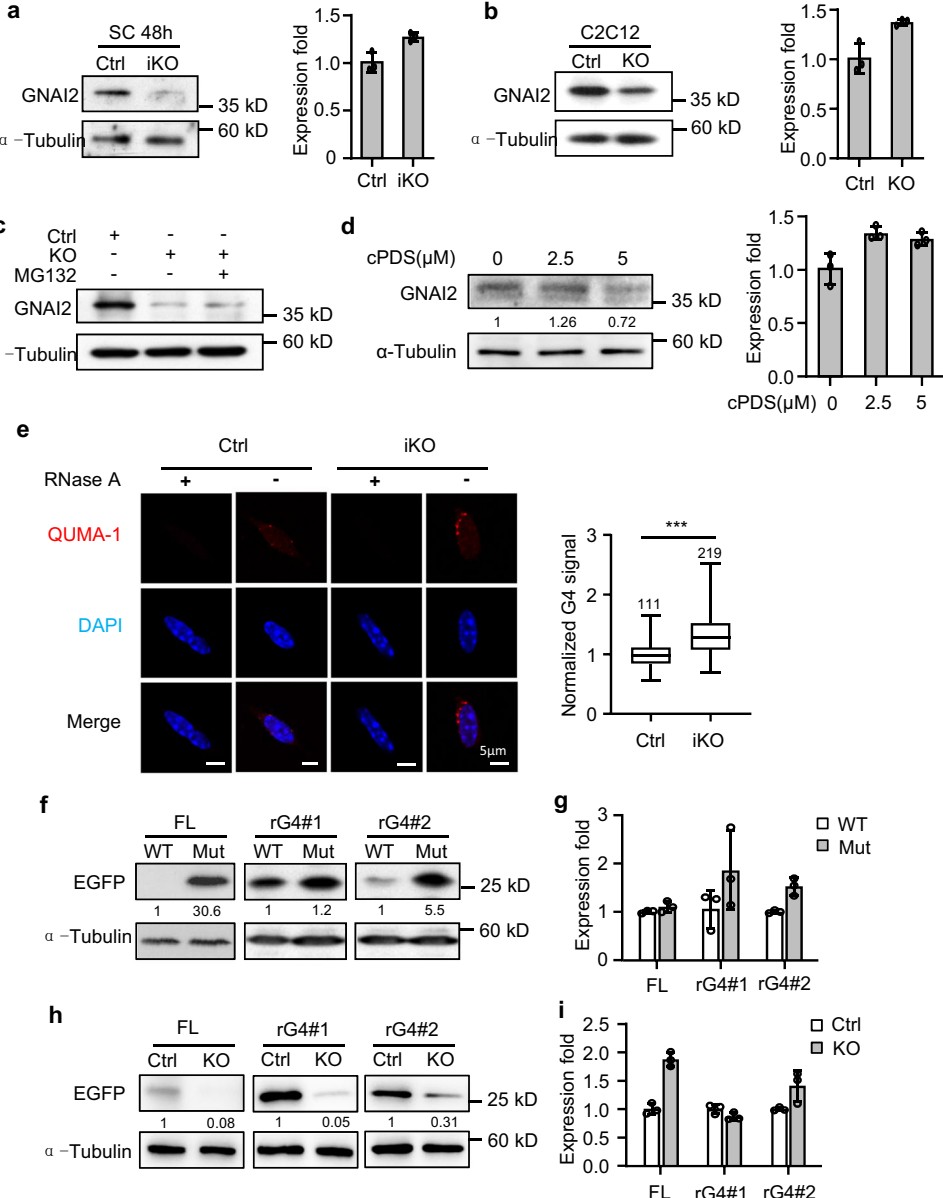

**Fig. 8 DHX36 promotes *Gnai2* translation by resolving the 5′ UTR rG4. a** Gnai2 protein and mRNA were detected in Ctrl and iKO SCs cultured for 48 h. **b** Gnai2 protein and mRNA were detected in proliferating Ctrl and Dhx36-KO C2C12 cells. **c** GNAI2 protein was detected in Ctrl, KO, or KO C2C12 cells treated with proteasome inhibitor MG132 for 8 h. **d** Left: SCs were treated with 2.5 or 5 μM cPDS and GNAI2 protein levels were detected by western blot. The relative intensity of each GNAI2 protein band normalized by α-Tubulin was calculated by ImageJ as the intensity for untreated cells set as 1. Right: *Gnai2* mRNA was detected by qRT-PCR in the above cells. *Gapdh mRNA* was used as a normalization control. **e** Left: rG4 formation in Ctrl and iKO SCs was detected by QUMA-1 staining. Cells treated with RNase A were used as negative controls. Scale bar = 5 μm. Right: Boxplot of normalized rG4 fluorescence signals which were calculated from the indicated number of cells from three pairs of Ctrl/iKO mice by Matlab R2014b using in-house scripts. The average intensity from Ctrl cells was set as 1. Horizontal lines represent median values. Error bars show the distribution from the minimum to the maximum data points. The boxes extend from the 25th to 75th percentiles. Significance was calculated by Student's *t* test (two-tailed unpaired), *P* = 3.1E-17. **f** EGFP reporter plasmids harboring WT or Mut full-length 5′ UTR, rG4#1, or rG4#2 were transfected into C2C12 cells and EGFP protein was detected by western blot. α-Tubulin was used as the internal loading control. **g** EGFP mRNA was detected by qRT-PCR from the above samples. **h** EGFP reporter plasmids harboring WT full-length 5′ UTR, rG4#1, or rG4#2 were transfected into Ctrl or Dhx36-KO C2C12 cells and EGFP protein was detected by western blot with α-Tubulin used as the internal loading control. **i** EGFP mRNA was detected by qRT-PCR from the above samples. Data represent the average of three independent experiments ± s.d. in **a**, **b**, **d**, **g**, and **i**. Source data are provided as a Source Data file.

significantly increased QUMA-1 signals in *Dhx36*-iKO vs Ctrl SCs (Fig. 8e), confirming the regulation of DHX36 on rG4 structures in SCs.

To further study whether DHX36 modulation of *Gnai2* mRNA translation is mediated by resolving the rG4 formation, we generated a EGFP reporter by inserting the WT or mutated full-length 5′ UTR of *Gnai2*, rG4#1, or rG4#2 sequence to the upstream of the EGFP ORF in the pEGFP-N1 plasmid[19] (Supplementary Fig. 8h, i). When transfected into C2C12 myoblasts, the protein levels of EGFP of all three WT reporters were drastically decreased compared with the corresponding mutated reporters as measured by western blot of EGFP

protein (Fig. 8f); notably, no significant changes in the levels of EGFP mRNA were observed (Fig. 8g), indicating that the rG4 formation in the *Gnai2* 5′ UTR indeed regulated mRNA translation. To strengthen this conclusion, the above WT reporters were transfected into Ctrl or *Dhx36*-KO cells; we found that the EGFP protein levels of all three reporters were dramatically downregulated in *Dhx36* KO vs Ctrl cells (Fig. 8h) while the mRNA levels remained unaltered (Fig. 8i).

Lastly, to test the functional connection of DHX36 and GNAI2 in SC proliferation, we examined SCs transfected SCs with a siRNA against *Gnai2* and a modest but significantly decreased level of EdU incorporation was observed (Fig. 9a, b). Conversely, when SCs were transduced with a *Gnai2*-expressing lentivirus, cell proliferation was promoted with a 26.9% increase in EdU+ SCs (Fig. 9c, d). These results thus suggest that DHX36 and GNAI2 conform to a regulatory axis for SC proliferation. In fact, transduction of SCs from *Dhx36* iKO with a *Gnai2*-expressing virus led to a 28.5% increase in EdU+ cells compared to SCs infected with Ctrl virus (Fig. 9e, f). Moreover, overexpression of the *Dhx36* plasmid also increased the GNAI2 protein level and rescued SC proliferation, with a 30.8% increase (Fig. 9g, h and Supplementary Fig. 10a, b). Altogether, the above results demonstrate that GNAI2 is indeed a downstream effector of DHX36 in promoting SC proliferation (Fig. 9i).

## Discussion

In this study, we demonstrate a fundamental role of DHX36, an RNA G-quadruplex (rG4) unwinding helicase, in skeletal muscle stem cells during muscle regeneration, via post-transcriptional regulation of mRNA metabolism. We found that specific inactivation of DHX36 in SCs impaired muscle regeneration by inhibiting their proliferative expansion. By combined CLIP-seq and polysome profiling, we discovered that DHX36 facilitated mRNA translation by binding and unwinding the rG4 structures formed at the 5′ UTR regions of target mRNAs. In particular, GNAI2 is a direct downstream effector of DHX36 proliferation-promoting functions. Beyond translational control, our analyses also revealed potential functions of DHX36 in regulating mRNA abundance as well as other aspects of mRNA metabolism. Our findings, in sum, uncover DHX36 as a versatile RBP and highlight the importance of post-transcriptional regulatory mechanisms in stem cells and tissue regeneration.

While epigenetic and transcriptional control of myogenesis and muscle regeneration has been studied extensively, the importance of post-transcriptional regulation in these processes is much less defined. Post-transcriptional regulation of gene expression permits cells to orchestrate rapid changes in protein levels from steady-state mRNAs, thus ensuring that a pool of primed stem cells is available to repair and maintain tissue function. Recent evidence supports the importance of post-transcriptional regulation in SCs. For example, some mRNAs are known to be transcribed and stored in quiescent SCs to allow the rapid production of protein products upon SC activation[9–11,49]. Our findings provide solid genetic evidence for this, demonstrating that DHX36 is an indispensable post-transcriptional regulator of skeletal muscle regeneration. Despite the increasing number of rG4-interacting RBPs discovered to date[50–53], loss of *Dhx36* dramatically impaired muscle regeneration with no compensation by other rG4-resolving helicases; this was confirmed by both constitutive and inducible inactivation of *Dhx36* in SCs in vivo. *Dhx36* loss caused a major defect in late SC activation and proliferation stages, which resulted in a drastic decrease in the cycling cell progeny required for muscle regeneration; this function was consistent with its expression dynamics, with DHX36 being induced as SCs became fully activated and began to proliferate

(Fig. 1b–d). Prior studies also showed the involvement of DHX36 in cellular proliferation, for example in erythrocytes[19,22–24]. Nevertheless, we cannot exclude that DHX6 may also regulate additional SC myogenic stages. It is also interesting to observe the puncta staining pattern of DHX36 protein in the cytoplasm of myoblast cells (Fig. 1e). Considering that DHX36 can be recruited to stress granules[54], a type of non-membrane-bound organelles formed through liquid–liquid phase separation (LLPS)[55,56], it will be interesting to explore in the future whether these puncta represent the formation of LLPS possibly on the translational regulatory sites as Sauer and co-workers[22] have suggested.

To elucidate the mechanisms underlying DHX36 function in SC biology, we integrated CLIP-seq binding and polysome profiling analyses. During the preparation of this manuscript, Sauer et al.[22] published the transcriptome-wide profiling of DHX36 in HEK293 cells by PAR-CLIP, using FLAG- or HA-tagged plasmids overexpressing DHX36. As the dose of RBP may significantly modulate a RNA structure, and thus the binding dynamics, obtaining the endogenous DHX36-binding profile (our study) is critical for understanding the physiological role of this helicase. As would be expected, a relatively smaller number of binding peaks (3032)/target mRNAs (1262) were identified in our study compared to what Sauer et al. [22] identified in HEK293 cells. Consistent with its almost exclusive cytoplasmic localization (Fig. 1e), it is not surprising the majority of DHX36 interacting transcripts are polyadenylated mRNAs, while a small portion arise from intergenic regions. Unsurprisingly, the highest binding density was located in the 5′ UTR regions although the highest number of binding sites was found in CDS and introns (Fig. 4e, f). We demonstrated that the dominant binding motifs in 5′ UTRs, 3′ UTRs, and CDS are all G-rich and predicted to form rG4s, strongly suggesting that the function of DHX36 is highly dependent on rG4 structures, at least in myoblasts. Interestingly, bulges and two-quartets, but not canonical rG4 structures were predicted, which suggests cell type-specific binding preference or dynamics for rG4 subtypes. In contrast to the study by Sauer and co-workers[22], which suggested that WT DHX36 cannot bind to AU-rich sequences while a mutant form with an inactive helicase (DHX36-E335A) can, we found that endogenous DHX36 in myoblasts interacts with 3′ UTR AREs in a large set of transcripts. When integrated with polysome profiling, we were able to demonstrate that 5′ UTR binding of DHX36 indeed confered a promoting function in translational efficiency, as a much higher number of DHX36-bound genes were down rather than upregulated. Nevertheless, we should point out that the number of mRNAs that interact with DHX36 through their 5′ UTRs and targeted for translational regulation is unexpectedly small (73). which could be due to the limited number of targets identified by our CLIP-seq (as mentioned above). DHX36 binding to the 3′ UTRs and CDS regions has not been previously documented. Notably, our data suggest that Dhx36 binding to rG4 in the 3′ UTRs has a role in translation. Interestingly, DHX36 binding at CDS had a marked effect on the TE of a number of transcripts; this effect may be rG4 independent despite the knowledge that rG4 formation in CDS can impact translational elongation[57–59]. Altogether, our findings uncover a wide range of DHX36 binding locations that may mediate diverse functional mRNA-regulatory mechanisms, which are either rG4 dependent or independent, and thus impact stem cell biology.

In addition to the translational effects, we explored the possible impact of DHX36 binding on mRNA abundance. Connecting rG4 binding to the mRNA abundance regulation revealed a high percentage of upregulated mRNAs with potential rG4-binding sites in their 5′ UTRs, suggesting that *Dhx36* loss increased transcript abundance; this could reflect a feedback result from retarded translation[60,61]. Intriguingly, we did not find any

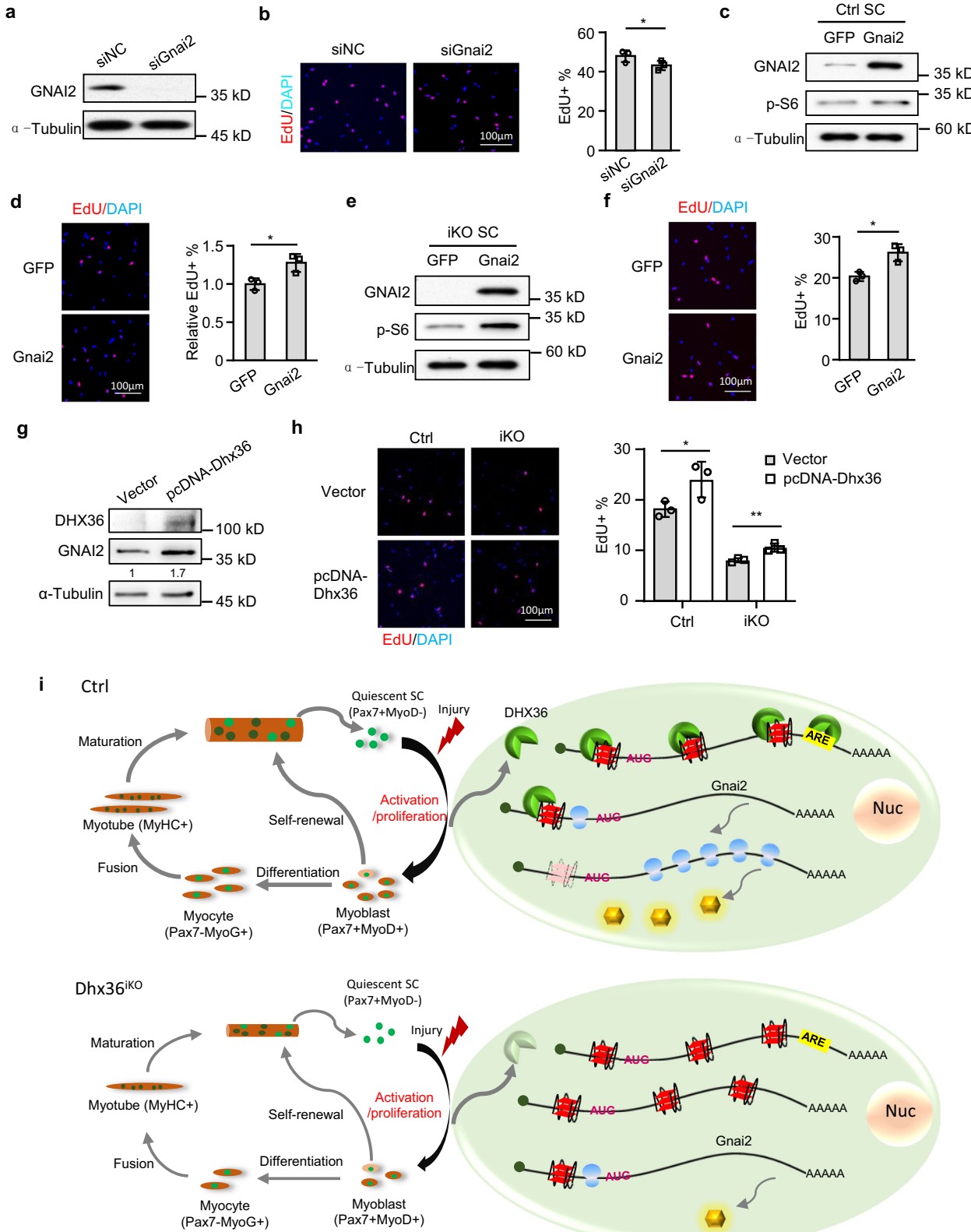

connection between mRNA abundance and Dhx36 binding to 3′ UTR AREs, despite the fact that DHX36 was first defined as a ARE-bound RBP that regulates mRNA degradation[12]; Sauer et al.[22] also failed to tease out whether ARE contributes to changes in mRNA abundance. The ambiguity of ARE in mediating DHX36 function thus needs to be further investigated. In addition, the rG4-dependent regulation of mRNA abundance by

DHX36 will be worth investigating. As rG4 could contribute to post-transcriptional processing and RNA decay, it is highly possible that DHX36 has diversified functional mechanisms through its connection with rG4.

Lastly, we proved that GNAI2 is a downstream effector of DHX36 in promoting SC proliferation. Minetti et al. [44] demonstrate that GNAI2 can promote SC proliferation and muscle

**Fig. 9 GNAI2 mediates DHX36 function in SC proliferation. a** C2C12 cells were transfected with siNC or siGnai2 oligos and the successful deletion of GNAI2 protein was detected by western blot with α-Tubulin as the internal loading control. **b** SCs were transfected with the above siRNA oligos for EdU assay. The percentage of EdU incorporation was quantified from 10 randomly selected fields/sample. Scale bar = 100 μm; $n = 3$ independent experiments. per group. **c** Ctrl SCs were infected with GFP or Gnai2-expressing lentivirus and the Gnai2 overexpression was confirmed by western blot. **d** EdU assay was conducted with the above cells and the relative EdU incorporation percentage was quantified from ten randomly selected fields per sample, with the value from GFP virus-infected cells set as 1. Representative images are shown. Scale bar = 100 μm; $n = 3$ mice per group. **e** iKO SCs were infected with GFP or Gnai2-expressing lentivirus and the expression of GNAI2 protein was confirmed by western blot at 48 h after infection. **f** EdU labeling was performed as above described and the incorporation percentage was quantified from ten randomly selected fields per sample. Representative images are shown. Scale bar = 100 μm; $n = 3$ mice per group. **g** Dhx36-KO myoblast cells were transfected with pcDNA3.1 vector plasmid or pcDNA3.1-mDhx36 overexpression plasmid for 48 h and the expression of Dhx36 and GNAI2 protein was confirmed by western blot. **h** Ctrl or Dhx36-iKO SCs were transfected with vector or pcDNA3.1-Dhx36 overexpression plasmid for 36 h and EdU was added to the cells 4 h before harvesting. The relative EdU incorporation percentage was quantified from ten randomly selected fields per sample. Scale bar = 100 μm; $n = 3$ mice per group. **i** Schematic illustration of the functional mechanism of DHX36 in regulating SC proliferation and muscle regeneration. Data represent mean ± s.d. (**b**, **d**, **f**, **h**) and Student's $t$ test (two-tailed unpaired) was used to calculate the statistical significance in **b**, **d**, **f**, **h** (iKO group); Student's $t$ test (two-tailed paired) was used to calculate the statistical significance in **h** (Ctrl group): *$P < 0.05$; **$P < 0.01$. Source data are provided as a Source Data file.

regeneration by both increasing the phosphorylation of GSK3β, p70S6K, and rpS6 as well as by suppressing the activity of HDACs. Consistently, we observed increased p-S6 level with GNAI2 overexpression (Fig. 9c, e). The mechanistic elucidation of the DHX36–GNAI2 regulatory axis in our study also serves as a proof-of-concept for dissecting how DHX36 binding to 5′ UTR rG4 promotes translational initiation. Various biophysical and biochemical methods were employed to demonstrate the presence of the rG4 and its binding with DHX36 as well as the regulation in translational initiation. We envision that many other targets, similar to *Gnai2*, exist in stem cells that mediate the pleiotropic DHX36 functions (see Supplementary Fig. 5e); this should be the subject of future intensive investigations.

## Methods

**Mice**. Pax7CreER (Pax7tm1(cre/ERT2)Gaka)[62] and Tg: Pax7-nGFP mouse strains[25] were kindly provided by Dr. Shahragim Tajbakhsh. Pax7Cre (Pax7tm1(cre)Mrc) mouse was kindly provided by Dr. Charles Keller. ROSAEYFP mouse was provided by Jackson Laboratory. Dhx36fl/fl strain was kindly provided by Dr. Zhongzhou Yang with the authorization of Dr. Yoshikuni Nagamine who originally generated this mouse strain[19,23]. Pax7Cre and Dhx36fl/fl mice were mated to generate Dhx36 conditional KO (Dhx36 cKO) mice (Ctrl: Pax7Cre/+; Dhx36+/+, cKO: Pax7Cre/+; Dhx36fl/fl). Pax7CreER mice were crossed with ROSAEYFP mice to generate the Pax7CreER; ROSAEYFP reporter mice. The Dhx36 inducible conditional KO (Dhx36 iKO) mice with EYFP reporter (Ctrl: Pax7CreER/+; ROSAEYFP/+; Dhx36+/+, iKO: Pax7CreER/+; ROSAEYFP/+; Dhx36fl/fl) were generated by crossing Pax7CreER; ROSAEYFP with Dhx36fl/fl mice. The mice were maintained in animal room with 12 h light/12 h dark cycles, 22–24 °C room temperature and 40–60% humidity at the animal facility in the Chinese University of Hong Kong (CUHK). All iKO animal handling procedures and protocols were approved by the Animal Experimentation Ethics Committee (AEEC) of CUHK (Ref. No. 17-013-MIS). All animal experiments with iKO mice followed the regulations and guidance of laboratory animals set in CUHK. The animal procedures involved cKO mice were supervised by the Ethical Committee of Animal Experimentation of the PRBB (CEEA-PRBB) and previously authorized by the corresponding Catalan Government committee, the Section for the Domestic Animal Protection, General Direction of Environmental and Nature Politics, Department of Territory and Sustainability.

**Animal procedures**. Inducible conditional deletion of Dhx36 was administered by injecting Tamoxifen (Tmx) (T5648, Sigma) intraperitoneally (IP) at 2 mg per 20 g body weight. For $BaCl_2$ induced muscle injury, ~2-month-old mice were intramuscularly injected with 50 μl of 10 μg/ml $BaCl_2$ solution into TA muscles and the muscles were harvested at designated time points for further analysis. For EdU incorporation assay in vivo, 2 days after $BaCl_2$ injection, EdU injection via IP at 0.25 mg per 20 g body weight was performed, followed by FACS isolation of SCs 12 h later. Cells were then collected and fixed with 4% PFA. EdU labeled cells were visualized using "click" chemistry with an Alexa Fluor® 594-conjugated azide. Images were captured with a fluorescence microscope (Leica). For cardiotoxin (CTX)-induced muscle regeneration, the TA, gastrocnemius, and quadriceps muscles were injected with CTX (Latoxan; 10−5 M). At the indicated time points (3- and 7-day post injury), mice were euthanized, and muscles were either snap frozen or dissected for satellite cell isolation by FACS.

**Grip-strength measurement**. Grip strength of all four limbs was measured using a grip-strength meter (Columbus Instruments, Columbus, OH). The animal was held so that all four limbs' paws grasped the specially designed mouse flat mesh assembly and the mouse was pulled back until their grip was broken. The force transducer retained the peak force reached when the animal's grip was broken, and this was recorded from a digital display. Five successful grip-strength measurements within 2 min were recorded. The maximum values were used for analysis. The mice were trained on the grip-strength meter before the trial. Maximal muscle strength was obtained as values of KGF (kilogram force) and represented in grams.

**Satellite cell sorting**. Skeletal muscle tissues from hindlimb and forelimb were dissected, gently minced with blades, and digested with Collagenase II (1000 U/ml) in Ham's F10 for 90 min in a shaking water bath at 37 °C. Digested tissue was then washed twice with rinsing media (10% HS, in Ham's F10) and centrifuged at 700 × g at 4 °C for 5 min. Second digestion was performed by adding Collagenase II (1000 U/ml) and Dispase (11 U/ml) in rinsing media and incubated in a shaking water bath at 37 °C for 30 min. Digested tissue was passed through a 20-gauge needle 12 times and filtered through a 40-μm filter followed by spinning at 700 × g for 5 min at 4 °C. SCs were then sorted out by FACS Aria Fusion (BD) and collected as GFP + groups. Isolated SCs were used either for RNA/protein extraction or were cultured in Ham's F10 medium supplemented with 20% FBS and bFGF (0.025 μg/ml) (growth medium).

**Single muscle fiber ex vivo culture**. Single myofibers were isolated from the extensor digitorum longus (EDL) muscles of 2–3-month-old mice by dissociation with collagenase II solution (800U/ml) at 37 °C in a water bath for 75 min. Dissociated single myofibers were manually collected and transferred to a new dish and the operation was repeated 2–3 times to remove dead fibers and debris. Isolated single myofibers were then either fixed in suspension immediately in 4% paraformaldehyde (PFA) for 15 min or maintained in suspension culture. For EdU incorporation assay, EdU was added to single myofibers which were cultured for 30 h and pulsed for another 4 h. Fibers were then fixed in 4% PFA for 15 min and stained following the EdU staining protocols provided by the manufacturer (Thermo Fisher Scientific, C10086).

**Cells**. Mouse C2C12 myoblast cells (CRL-1772) and 293T cells (CRL-3216) were obtained from American Type Culture Collection (ATCC) and cultured in DMEM medium with 10% fetal bovine serum, 100 units/ml of penicillin, and 100 μg of streptomycin (growth medium, or GM) at 37 °C in 5% $CO_2$. For in vitro differentiation, C2C12 myoblasts were plated and cultured to high confluence and changed to the differentiation media (DM) (DMEM with 2% horse serum). Cells were cultured in DM and harvested for western blot on days 1, 3, and 5. WT primary myoblast cells were cultured in Ham's F10 medium supplemented with 20% FBS and bFGF (0.025 mg/ml) (growth media, GM). For in vitro differentiation experiments, myoblasts were plated and cultured to high confluence and changed to DMEM with 5% goat serum (differentiation media, DM). Cells were cultured in DM for 24, 48, 72 or 96 h. For generating Dhx36 knockout (KO) C2C12 cells, two sgRNAs expressing plasmids[63] which targeted the third exon of Dhx36 were transfected to C2C12 and clones were selected by genotyping. Lentiviruses expressing GFP or Gnai2 were packaged in 293T cells as previously described[64]. Ctrl or iKO SCs were infected by GFP or Gnai2 expression virus after attaching to the culture plate. Forty-eight hours after, infection cells were used for EdU incorporation assay, collected for western blot or qRT-PCR. For DHX36 overexpression, pcDNA3.1 vector plasmid or pcDNA3.1-mDhx36 plasmid was transfected to SCs or C2C12 cells. Forty-eight hours after, the transfection cells were used for EdU incorporation assay, collected for western blot or qRT-PCR. For

cPDS treatment, SCs or C2C12 cells were treated with DMSO, 2.5 μM or 5 μM cPDS for 24 h then used for EdU incorporation assay, collected for western blot or qRT-PCR.

**Preparation of DNA oligonucleotides**. All DNA oligonucleotides were purchased from Integrated DNA Technologies (IDT). The Cy5 fluorescence-labeled DNAs were purified by HPLC. The quality of each oligonucleotide was confirmed by ESI-MS by IDT, with a single peak at the expected size, thus all DNA oligonucleotides were used without further purification. The sequences of the oligos can be found in Supplementary Dataset 4.

**Plasmids**. Two sgRNAs for generating *Dhx36* knockout (KO) C2C12 cells were designed by CRISPOR[63] to target the third exon of *Dhx36* and cloned into PX458 vector at the BbsI site. For lentivirus package and infection, mouse Gnai2 ORF was amplified from C2C12 cell cDNA and cloned into pLenti through PmeI and EcoRI restriction sites. For generating DHX36 overexpression plasmid, mouse DHX36 ORF was amplified from C2C12 cell cDNAs and cloned into pcDNA3.1 vector through KpnI and XhoI restriction sites. For EGFP reporter assays, the wild-type full-length 5′ UTR of Gnai2 mRNA or the mutated one with all the GGG mutated to GAG were cloned to the pEGFP-N1 vector at the XhoI/EcoRI sites. WT or Mut rG4#1 or rG4#2 sequenced were synthesized by BGI Genomics and cloned directly to pEGFP-N1 vector. The sequences of the cloning primers and oligos are shown in Supplementary Dataset 4.

**EGFP reporter assay**. The EGFP reporter assay was conducted as previously reported[19]. Briefly, reporter vectors were transfected to the C2C12 cell lines and cultured for 48 h before being harvested for RNA and protein extraction. Western blot and qRT-PCR were used to evaluate the protein and mRNA level of EGFP, respectively.

**Real-time PCR**. Total RNAs from SCs or C2C12 cells were extracted using TRIzol reagent (Life Technologies) according to the manufacturer's protocol. QRT-PCR was performed by using SYBR Green Master Mix (Applied Biosystem) on ABI PRISM 7900HT (Applied Biosystem). *18s rRNA* or *GAPDH* mRNA was used for normalization. All the experiments were designed in triplicates. Primers used for qRT-PCR are shown in Supplementary Dataset 4.

**RNA pull-down assay**. RNA pull-down assay was conducted as previously described[65,66]. Briefly, biotin-labeled RNAs were in vitro transcribed with the Biotin RNA Labeling Mix (Roche) and T7/T3 RNA in vitro transcription kit (Ambion) and purified with RNAeasy kit (Qiagen). C2C12 cells were harvested and lysed with RIPA buffer. Biotin-labeled RNAs were denatured at 90 °C for 2 min and then renatured with RNA structure buffer (10 mM Tris pH 7, 0.1 M KCl, 10 mM MgCl₂) at RT for 20 min. Pre-folded RNA was then mixed with cell lysate and incubated at RT for 1 h. Streptavidin agarose beads (Invitrogen) were then added to the binding mix and incubated for another hour. Beads were washed in Handee spin columns (Pierce) for five times using RIPA buffer (50 mM Tris–HCl, pH 7.5, 150 mM NaCl, 1.0 mM EDTA, 0.1% SDS, 1% sodium deoxycholate, and 1% Triton X-100) and binding proteins were retrieved from beads and used for further analysis by running 10% SDS–PAGE gel for western blot analysis.

**RNA immunoprecipitation**. RNA immunoprecipitation assay was conducted as previously described[65,67]. C2C12 myoblast cells were treated with Ctrl (same volume of DMSO in the culture medium as cPDS) or 5 μM cPDS for 24 h and collected for RIP. Cells were crosslinked in medium with 1% formaldehyde at 37 ºC for 10 min followed by adding 0.125 M of glycine to quench formaldehyde. Cells were then washed with PBS for three times and lysed with RIPA buffer (50 mM Tris, pH 7.4, 150 mM NaCl, 1 mM EDTA, 0.1% SDS, 1% NP-40, 0.5% sodium deoxycholate, 0.5 mM DTT, 1/100 volume of RNase-out and 1x protease inhibitor cocktail) at 4 ºC for 30 min. The supernatant was collected after centrifugation at 13,000 × *g* at 4 ºC for 10 min. In total, 2 μg of antibodies against Dhx36 (Abcam, ab70269) or isotype IgG (Santa Cruz Biotechnology) were incubated with same volume of cell lysate at 4 °C for 2–4 h followed by adding 50 μl of Protein G Dynabeads™ (ThermoFisher Scientific) to the lysate with further incubation at 4 °C overnight. Wash beads at 4 °C with RIPA buffer for four times and then add Proteinase K to the beads with incubation at 45 °C for an hour. After incubation, TRIzol was added (ThermoFisher Scientific) to the sample for RNA extraction. Extracted RNA were resuspended in 20 μl of RNase-free water and cDNAs were obtained from reverse transcription. QRT-PCR was performed with cDNAs by using SYBR Green Master Mix (Applied Biosystems). Relative enrichment was calculated as the amount of amplified DNA normalized to the values obtained from IgG immunoprecipitation which was set as 1.

**Preparation of in vitro-transcribed RNAs**. In vitro-transcribed (IVT) RNAs were made using pre-annealed DNA hemiduplex and HiScribe™ T7 High Yield RNA Synthesis Kit following the manufacturer's protocol. Each RNA was purified using the 7 M urea, 15% denaturing acrylamide gel (Life Technologies), and the desired RNA gel band was sliced under UV. The gel piece was crushed and soaked in 1×

10 mM Tris pH 7.5, 1 mM EDTA, 800 mM LiCl (1× TEL800). The mixture was under constant shaking at 1300 rpm overnight at 4 °C. Next, the mixture was filtered against 0.22-μm filter and purified using RNA Clean & Concentrator-5 Kit (Zymo). The IVT RNA was then stored at −20 °C before use.

**Reverse-transcriptase stalling assay**. Reverse-transcriptase stalling assay was performed similar to what was reported previously[46]. Briefly, 3–5 pmol of IVT RNA was added up to 4.5 μL with nuclease-free water, and 1 μL of 5 μM Cy5 fluorescence-labeled was added. The mixture was heated at 75 °C for 3 min, followed by incubation at 35 °C for 5 min. At the beginning of the 35 °C incubation, 3 μL of reverse transcription buffer was added to reach a final concentration of 150 mM KCl, 4 mM MgCl₂, 20 mM Tris pH 7.5, 1 mM DTT, and 0.5 mM dNTPs. For cation-dependent experiments, either 150 mM KCl or LiCl was used, unless otherwise stated. For ligand-dependent experiments, 1 μL of 20 μM PDS was added after the reverse transcription buffer. The 9.5 μL of mixture was heated up to 55 °C and 0.5 μL of Superscript III (200U/μL) was added to make up the 10 μL reaction. The reverse transcription mixture was incubated at 55 °C for 15 min, and then 0.5 μL of 2 M NaOH was added at the end of the step. Then the mixture was incubated at 95 °C for 10 min to inactivate the SSIII and degrade the RNA template. For RTS, 10 μL of 2× stopping dye solution which contains 20 mM Tris, pH 7.5, 20 mM EDTA, 94% deionized formamide was added to the reaction mixture. Orange G dye was added as tracker.

**Fluorescence spectroscopy**. Fluorescence spectroscopy was performed similar to what was reported previously[68,69]. Briefly, 1 μM RNAs in a reaction volume of 100 μL were mixed in 10 mM LiCac (pH 7.0) buffer and 150 mM KCl or LiCl. Samples were annealed at 95 °C for 5 min and cooled to room temperature for 15 min for renaturation. 5 μL of 20 μM NMM or ThT were added to the samples and samples were excited at 394 nm and 425 nm for NMM and ThT ligands, respectively. The emission spectrum was collected from 550 to 750 nm for NMM and 440 to 700 nm for ThT. 1-cm path-length quartz cuvette was used and measurements were performed using HORIBA FluoroMax-4. Spectra were acquired every 2 nm at 25 °C for WT and Mut samples. The entrance and exit slits were 5 and 2 nm, respectively.

**CD spectroscopy**. CD spectroscopy was performed using a Jasco CD J-150 spectrometer with a 1 cm path-length quartz cuvette. The oligos consisting of rG4 sequences were prepared in 10 mM LiCac buffer (pH 7.0) and 150 mM KCl/ LiCl. The mixtures, with a reaction volume of 2 mL, were then vortexed and heated at 95 °C for 5 min and allowed to cool to room temperature. The oligos were examined from 220 to 310 nm at a 2 nm interval, and the data were blanked and normalized to mean residue ellipticity. The data were interpreted using Spectra Manager Suite (Jasco Software).

**UV melting assay**. UV melting spectra was carried out using a Cary 100 UV-vis spectrophotometer with a 1-cm path-length quartz cuvette. In total, 2 mL of samples containing oligos with rG4 sequences were prepared in 10 mM LiCac buffer (pH 7.0) and 150 mM KCl. The samples were vortexed and heated at 95 °C for 5 min and allowed to cool to room temperature. The samples were then transferred into the cuvettes and sealed using Teflon tape. The samples were examined at 295 nm from 5 °C to 95 °C with 0.5 °C increment per minute. The data were blanked and smoothed over ten data points.

**Immunoblotting and immunofluorescence**. For western blot assay, in vitro cultured cells were harvested, washed with ice-cold PBS and lysed in RIPA buffer supplemented with protease inhibitor cocktail (88266, Thermo Fisher Scientific) for 20 min on ice. Whole cell lysates were subjected to SDS–PAGE and protein expression was visualized using an enhanced chemiluminescence detection system (GE Healthcare, Little Chalfont, UK) as described before[31]. The following dilutions were used for each antibody: DHX36 (Abcam ab70269; 1:5,000), α-Tubulin (Santa Cruz Biotechnology sc-23948; 1:5000), MyoD (Santa Cruz Biotechnology sc-760; 1:1000), GNAI2 (Abcam ab157204; 1:5,000), CCND1 (Santa Cruz Biotechnology sc-718; 1:5000), CCNA1 (Santa Cruz Biotechnology sc-596; 1:5000), p-S6 (Cell signaling technology #4858; 1:5000), HnrnpL (Santa Cruz Biotechnology sc-28726; 1:5000), DHX9 (Santa Cruz Biotechnology sc-137232; 1:5000). For immunofluorescence staining, cultured cells and myofibers were fixed in 4% PFA for 15 min and permeabilized with 0.5% NP40 or 0.2% Triton X-100 in PBS followed by block with 3% BSA in PBS for 1 h. Primary antibodies were applied to samples with indicated dilution below and the samples were kept at 4 °C overnight. For Dhx36 staining, FACS-sorted satellite cells were plated in 15-well chambers (IBIDI) and cultured in Ham's F10 supplemented with 20% FBS and bFGF (0.025 μg/ml). For quiescence time point, cells were directly fixed with PFA 4% for 10 min; for activation time points, cells were cultured for 24 h and fixed with PFA 4%. Cells were incubated with rabbit anti-DHX36 (Proteintech 13159-1-AP) primary antibody after blocking for 1 h at room temperature with 3% BSA (Sigma) in PBS. Cells were then washed with PBS and incubated with secondary antibody conjugated to AF488 or AF555 fluorochromes, and nuclei were stained with DAPI (Invitrogen) for 2 h at room temperature. After washing, Fluoromount (Thermo Fisher) was added to the wells of the chamber to preserve the fluorescence. For Pax7 staining on frozen muscle sections, an antigen retrieval step was performed before blocking

through boiling samples in 0.01 M citric acid (pH 6.0) for 5 min in a microwave. After 4% BBBSA (4% IgG-free BSA in PBS; Jackson, ref: 001-000-162) blocking, the sections were further blocked with the Donkey anti-Mouse IgG (H+L) (1/100 in PBS; Jackson, ref: 115-007-003) for 30 min. The biotin conjugated anti-mouse IgG (1:500 in 4% BBBSA, Jackson, ref: 205) and Cy3-Streptavidin (1:1250 in 4% BBBSA, Jackson, ref: 016–160) were used as secondary antibodies. H&E staining on frozen muscle sections was performed as previously described[31]. All fluorescent images were captured with a fluorescence microscope (Leica DM6000B) by Leica LAS AF software (LAS AF2.6.3). Primary antibodies used include: MyoD (Santa Cruz Biotechnology sc-760; 1:200) and Myogenin (Santa Cruz Biotechnology sc-576; 1:200); Pax7 (Developmental Studies Hybridoma Bank; 1:50) and MF20 (Developmental Studies Hybridoma Bank; 1:50); eMyHC (Leica NCL-MHC-d; 1:200) and laminin (Sigma-Aldrich L9393; 1:800); Ki67 (Santa Cruz Biotechnology sc-15402; 1:200); MyoD (Dako M3512; 1:200) for staining of muscle cryosections. Confocal images of isolated satellite cells were taken using a Zeiss LSM-780 confocal system with a Plan-Apochromat 63× /1.4 NA oil. Acquisition was performed using Zeiss LSM software Zen Black. Images were slightly modified with ImageJ in which background was reduced using background subtraction and brightness and contrast were adjusted.

**In cellulo rG4 staining**. SCs were fixed with 4% paraformaldehyde in PBS for 15 min at room temperature. The fixed cells were then stained with 1 µM probe QUMA-1 for 20 min at 37 °C. For cells treated with RNase A, fixed cells were firstly permeabilized with 0.5% Triton X-100 in PBS at RT for 20 min followed by incubation with 100 µg/ml RNase A for 1 h at 37 °C and staining with 1 µM probe QUMA-1 for 20 min at 37 °C. After incubation with QUMA-1, SCs cells were washed three times with PBS and confocal images were taken using a Zeiss LSM-780 confocal system.

**RNA-seq and data analysis**. Total RNA of quiescent satellite cells (SC$_{T0}$) isolated after in situ fixation, freshly isolated SCs without prior fixation (SC$_{T8}$) (fixed after isolation) and SCs cultured for 24, 48, or 72 h were collected and subjected to total RNA-seq. RNA libraries were generated and sequenced with DNBseq platform by BGI, Hong Kong. In brief, rRNAs were first depleted from total RNAs and the remaining RNAs were fragmented and then reverse transcribed. The resulting DNA fragments were end-repaired, A-nucleotide added and adaptors ligated. The resulting libraries were heated to seperate into ssDNA and further cyclized to make DNA nanoballs (DNBs). DNBs were sequenced by DNBSEQ-G400 sequencer in BGI-Hong Kong. For the data analysis, the adapter sequences and the low-quality bases were trimmed from 3′ to 5′ ends for each read and the reads that shorter than 50 bp were then dumped. The reads that passed the quality control were further mapped to mouse genome (mm9) with TopHat2[70]. Cufflinks[71] were then used to estimate transcript abundance of RNA-Seq experiments in fragments per kilobase per million (FPKM).

**CLIP-seq and data analysis**. C2C12 cells in growth medium were crosslinked under UV and then lysed for immunoprecipitation using antibody against DHX36 as described previously[32]. C2C12 cells in growth medium were washed with PBS and crosslinked by irradiation one time at 120 mJ/cm$^2$ on ice for 1–1.5 min. The cells were then collected and flash-frozen in liquid nitrogen and stored at −80 °C. Cells were lysed in 0.6 ml Lysis Buffer (1× PBS with 0.1% SDS, 0.5% deoxycholate, and 0.5% NP-40) supplemented with 1 mM DTT, 30 µl of RNasin (Takara) and 60 µl of RQ1 DNase (Promega) at 37 °C for 2 min on a thermomixer. Then stand tubes on ice for 5 min and centrifuge at 13,000 × g for 20 min at 4 °C. Collect the supernatant and add protein A Dynabeads to the lysate which were pre-incubated with DHX36 antibody at RT for 1 h. Rotate beads/lysate mix for 1 h at 4 °C. Wash beads two times with Lysis Buffer, two times with high-salt wash buffer (5× PBS with 0.1% SDS, 0.5% deoxycholate and 0.5% NP-40) and two times with PNK Buffer (50 mM Tris–Cl, pH 7.4, 10 mM MgCl$_2$ and 0.5% NP-40). Fragmentate RNA tag with MNase at 37 °C for 10 min followed by two times wash with PNK/EGTA Buffer (50 mM Tris-Cl, pH 7.4, 20 mM EGTA, 0.5% NP-40), two times with Lysis Buffer, and two times with PNK Buffer. Then treat the beads with alkaline phosphatase at 37 °C for 10 min. Then on-bead 3′ RNA Linker ligation mix was incubated at 16 °C overnight followed by three times wash with PNK Buffer. Then the beads were treated with PNK mix at 37 °C for 10 min followed by three times wash with PNK Buffer. Resuspend beads in 30 µl of PNK Buffer plus 30 µl of Novex loading buffer and incubate at 70 °C for 10 min, using the supernatant for loading to the 10% Bis–Tris gel. After the gel running, transfer the gel to nitrocellulose membrane. RNA-protein complex with 15–20 kD larger than the protein MW were retrieved for library preparation and then sequenced with Illumina sequencing system. The first 6 nt of CLIP-seq raw reads were removed due to their low quality reported by FastQC[72], adaptors were further trimmed using Trimmomatic[73] with the minimal length threshold equal to 18 nt. After duplicates were removed, alignment was performed against mouse genome MM9 using Bowtie2[74] by allowing two insertions or deletions, two mismatches and only reads with greater than 90% identity were saved. Peak calling on the eligible reads was conducted via Piranha[75] using bin size parameter equal to 100 nt and significant threshold of 0.05. Reproducibility between biological replicates was examined by both Pearson Correlation Coefficient calculated using bam file coverages within 10-

kbp bin size from deepTools multiBamSummary module[76] and the proportion of peak overlapping using bedtools[77]. Genomic distribution of the peaks was annotated using mouse MM9 genome by HOMER[78] and in-house scripts.

To perform the de novo motif discovery on DHX36-binding sites, MEME[33] was employed using -rna option, -nmotifs equal to 20 and 18 nt motif width, consistent with the length threshold used in CLIP-seq data preprocessing. The motif discovery on intronic binding sites was conducted on those sites not overlapping with intronic repeat elements. The most significant motifs and the corresponding E-values were reported. As for the identification of sequences similar with the 18 nt de novo motifs, FIMO[79] with "-norc -parse-genomic-coord" parameters was used. To predict the potential rG4 forming sites in DHX36-binding regions, we searched for the four subtypes defined as follows: $G_3L_{1-7}$, canonical rG4s with loop length between 1–7 nt (" $(G_{3+}N_{1-7})\{3,\}cG_{3+}$ ", with $N$ = A, U, C, or G); long loops, rG4s with any loop of length >7 nt, up to 12 nt for lateral loops and 21 nt for the central loop (e.g., " $G_{3+}N_{8-12}G_{3+}N_{1-7}G_{3+}N_{1-7}G_{3+}$ " or " $G_{3+}N_{1-7}G_{3+}N_{13-21}G_{3+}N_{1-7}G_{3+}$ "); bulges, rG4s with a bulge of 1–7 nt in one G tract, or multiple 1 nt bulges (e.g., " $G_{3+}N_{1-9}G_{3+}N_{1-9}(GGH_{1-7}G|GH_{1-7}GG)N_{1-9}G_{3+}$ " or " $(GGHG|GHGG)N_{1-9} (GGHG|GHGG)N_{1-9}G_{3+}N_{1-9}G_{3+}$ ", with H = A, U, or C); 2 quartet, rG4s with four tracts of two consecutive Gs (" $(G_{2+}N_{1-9})\{3,\}G_{2+}$ "); G ≥50%, sequences that contain more than 50% G content and do not fall into the four previous categories; others, not in any previous category. When matching multiple categories, a region was assigned to the type with highest predicted stability, i.e. (from first to last), canonical rG4s, long loops, bulges, and 2 quartet. When predicting rG4 formation in the 18 nt CLIP-seq motif-enriched regions in Fig. 4k–m, only the rG4 sites overlapped with more than 9nt of the motif were used for the hierarchical assignment of rG4 subtypes described above.

To predict de novo structural motif, RNAshapes[40] tool was used to annotate the six generic shapes including Stems (S), Multiloops (M), Hairpins (H), Internal loops (I), dangling end (T), and dangling start (F) in RNA sequences. The structural motif logos were generated using WebLogo3 webserver[80].

To infer potential RBP binding in CDS sites, 8-nt motifs were compared with known RBP binding motifs from ATtRACT[81] database. Briefly, Position Weight Matrix (PWM) of all RBPs were downloaded from ATtRACT and converted to minimal MEME motif format by in-house scripts. Tomtom tool from the MEME suite was then utilized to compute the similarity between the 8-nt motifs and motifs in ATtRACT, with minimum overlapping threshold set to 5 nt and significance threshold equal to 3.0e-4.

To identify potential C/D box methylation guide snoRNAs, snoSCAN[41] was used with default parameters and settings. The Dhx36 intronic binding sites overlapped with neither annotated snoRNAs (information from HOMER software) nor intronic repeat elements (information downloaded from UCSC table browser[82]) were used as inputs for snoSCAN.

To identify the AU-rich element (ARE) in Dhx36-binding sites, we firstly searched for sequences matching three classes of conventional ARE patterns. The class one, defined as "several dispersed AUUUA motifs in an U-rich context", included sequences matching the pattern of cluster 1 defined in ref. [83]. We also added sequences matching "WWUUUWW", "WWWUUUWWW", and "WWWWUUUUWWWW" to class one, where W is A or U and these patterns were used by the ARE database AREsite2[84]. The definition of class two was identical with cluster 2, 3, 4, 5 from ref. [83], which includes AREs with two to five overlapping AUUUA motifs. The class three was defined as "U-rich regions"[39] so sequences with more than 5 continuous "U"s were collected. When matching multiple categories, the shorter AREs were removed. Finally, closely located sequences (within 5 nt) in classes one and three were merged together with bedtools merge, to produce the final ARE set.

To calculate the Minimum Free Energies (MFEs) of RNA secondary structures, RNAfold program from ViennaRNA 2.4.10[85] was used. The MFEs of dsRNA secondary structures were calculated at 37 °C with parameters "–MEA -p0 -d2–noLP" and named as $\triangle G^0_{dsRNA}$ while MFEs for rG4 structures were calculated using the following formula:

$$\triangle G^0_{rG4} = \triangle G^0_{dsRNA} - \triangle G^0_{dsRNA+rG4} \qquad (1)$$

where $\triangle G^0_{dsRNA+rG4}$ were the MFEs calculated when considering rG4 formation into RNAfold prediction. Both MFE values were normalized by the input sequence length and the types of input sequence were the 5′ UTR regions and the ARE sites with a 30 nt extension on both sides. For all the MFE comparisons, we randomly selected 1000 genes without Dhx36 bindings and showed no TE or mRNA changes as the background set.

**Polysome profiling and data analysis**. Ctrl or Dhx36-KO C2C12 cells were cultured in growth medium and collected for polysome profiling as previously described[86]. Prior to lysis, cells were treated with cycloheximide (100 µg/ml) for 10 min at 37 °C. Cells were then washed with ice-cold PBS supplemented with 100 µg/ml cycloheximide and further lysed in 300 µl of lysis buffer (10 mM HEPES pH 7.4, 150 mM KCl, 10 mM MgCl$_2$, 1% NP-40, 0.5 mM DTT, 100 µg/ml cycloheximide). After lysing the cells by passing eight times through 26-gauge needle, the nuclei and the membrane debris were removed by centrifugation (15,682 × g, 10 min, 4 °C). The supernatant was then layered onto a 10-ml linear sucrose gradient (10–50% [w/v], supplemented with 10 mM HEPES pH 7.4, 150 mM KCl, 10 mM MgCl$_2$, 0.5 mM DTT, 100 µg/ml cycloheximide) and centrifuged (160,000 × g, 120 min, 4 °C)

in an SW41Ti rotor (Beckman). After centrifugation, the sediment in the tube went through a UV detector at the wavelength of 254 nm and simultaneously the required fractions (≥5 ribosomes, polysome) were collected and polysome-associated RNAs were extracted by Trizol-LS. TruSeq Stranded Total RNA libraries were prepared with 500 ng RNA according to the manufacturer's protocol (Illumina). The libraries were sequenced in 2 × 100 nt manner on HiSeq 2000 platform (Illumina).

The sequenced total cytoplasmic mRNA and polysome-associated RNA libraries were quantified following the RNA-seq processing procedures. Briefly, after the adaptor trimming, quality filtering and duplication removal using in-house scripts[87], the sequenced fragments were mapped to reference mouse genome (MM9) using TopHat2[70]. Cufflinks[88] was then used to derive the fragments per kilobase per million (FPKM) values. TE values were then calculated as the ratio between the abundance of polysome-associated RNAs (FPKM-polysome) and total cytoplasmic mRNAs (FPKM-total) in Ctrl and Dhx36-KO conditions. The fold change of TE upon DHX36 deletion was further calculated as the TE values in KO divided by TEs in Ctrl cells, generating $\log2(\Delta TE)$. Only those genes with TE up- or downregulated in both replicates, were used for further analysis with their averaged FPKM-polysome and FPKM-total values. When intercepting with CLIP-seq data to dissect the effect of DHX36 binding on TE and total mRNA abundance, transcripts with averaged FPKM-total values larger than 1 in either Ctrl or KO cells were used.

**Gene Ontology analysis.** ClusterProfiler[89] was used for the Gene Ontology (GO) analysis with Entrez gene IDs converted from DAVID[90] tool as inputs. The adjusted P or P values were reported with the GO terms.

**Statistics and reproducibility.** Data represent the average of at least three independent experiments or mice ± s.d. unless indicated. The statistical significance of experimental data was calculated by the Student's t test (two-sided). $*P < 0.05$, $**P < 0.01$, $***P < 0.001$ and n.s.: not significant ($P \geq 0.05$). The statistical significance for the assays conducted with SCs from the same mouse with different treatment was calculated by the Student's t test (paired). $*P < 0.05$, $**P < 0.01$, $***P < 0.001$ and n.s.: not significant ($P \geq 0.05$). The statistical significance of minimal free energy (MFEs) between groups in Figs. 5o, 6i, and 6k was assessed using one-tailed Mann-Whitney test. $***P < 0.001$, and n.s.: not significant ($P \geq 0.05$). The P values of DHX36 binding sites in Supplementary Dataset 1 were assigned by the Piranha peak caller software[75]. Specifically, a single zero-truncated negative binomial distribution was fit to the input data and each region was assigned a P value based on the fitted distribution. Representative images of at least three independent experiments are shown in Fig. 1c, e, g, k; 2c, d, e; 3j; 7c, f; 8d, f, h; 9a, c, e, g; and Supplementary Fig. 1c; 2d, f; 9a. Representative images of two independent experiments are shown in Figs. 4a and 7e. RNA-seq data of one experiment are shown in Fig. 1b and Supplementary Fig. 1b.

**Reporting summary.** Further information on research design is available in the Nature Research Reporting Summary linked to this article.

## Data availability

RNA-seq data of quiescent satellite cells ($SC_{T0}$) isolated after in situ fixation, freshly isolated SCs without prior fixation ($SC_{T8}$) (fixed after isolation) and SCs cultured for 24, 48, or 72 h are deposited in Gene Expression Omnibus (GEO) database with the accession codes GSE175501. CLIP-seq and Polysome profiling data are deposited in GEO database with the accession codes GSE151124. The data supporting the findings of this study are available from the corresponding author on reasonable request. Source data are provided with this paper.

## Code availability

The code for rG4 site and AU-rich element identification has been deposited in a Github repository with the DOI identifier as 10.5281/zenodo.4732879 (https://github.com/jieyuanCUHK/DHX36_paper).

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

## Acknowledgements

We thank Prof. Zhongzhou Yang for his generous sharing of the *Dhx36^{fl/fl}* mouse and the pEGFP-N1 plasmid. This work was supported by General Research Funds (GRF) from the Research Grants Council (RGC) of the Hong Kong Special Administrative Region (14115319, 14100018, 14100620, 14106117 and 14106521 to H.W.; 14120420, 14116918, and 14120619 to H.S.); Guangdong Natural Science Foundation from Guangdong Basic and Applied Basic Research Foundation to X.C. (Project code: 2019A1515010670); the National Natural Science Foundation of China (NSFC) to H.W. (Project code: 31871304); Collaborative Research Fund (CRF) from RGC to H.W. (C6018-19GF);

CUHK Direct Grant for Research to H.W. (Project code: 4054482); NSFC/RGC Joint Research Scheme to H.S. (Project code: N_CUHK 413/18); Hong Kong Epigenomics Project (EpiHK) Fund to H.W. and H.S.; Area of Excellence Scheme (AoE) from RGC (Project number: AoE/M-402/20). Work in PMC laboratory was supported by Spanish Ministry of Science, Innovation and Universities, Spain (grants RTI2018-096068-B-I00 and SAF 2015-70270-REDT, a María de Maeztu Unit of Excellence award to UPF [MDM-2014-0370], and a Severo Ochoa Center of Excellence award to the CNIC [SEV-2015-0505]), ERC-2016-AdG-741966, La Caixa-HEALTH (HR17-00040), MDA, UPGRADE-H2020-825825, AFM and DPP-E. S.C. is recipient of a FI fellowship from AGAUR. Work in CKK laboratory was supported by Shenzhen Basic Research Project (JCYJ20180507181642811), Research Grants Council of the Hong Kong SAR (CityU 11101519, CityU 11100218, N_CityU110/17, CityU 21302317), Croucher Foundation Project (9509003, 9500030), State Key Laboratory of Marine Pollution Director Discretionary Fund, City University of Hong Kong projects (6000711, 7005503, 9680261, and 9667222 to C.K.K.). Work in YX laboratory was supported by the National Natural Science Foundation of China (32025008 and 91940306 to Y.X.).

## Author contributions

X.C. and H.W. designed the iKO mouse and molecular mechanism experiments; X.C. and G.X. conducted the experiments; Y.Z., L.H., and Y.L. provided technical supports; J.Y. analyzed the NGS data; S.C.S., J.I., C.M., and E.P. performed Dhx36 cKO mouse and part of iKO mouse analyses; W.W. and W.C. contributed to Polysome profiling; X.M. conducted RTS gel assay, CD and UV melting assay; S.W.L. and M.I.U. conducted ThT and NMM staining; Y.X. supervised the CLIP-seq, D.W., L.W., and X.Y. conducted CLIP-seq; Y.N. generated the *Dhx36^{fl/fl}* mouse; C.K.K. supervised rG4 analyses; H.S. supervised computational analyses; P.M.C. supervised Dhx36 cKO mouse and part of iKO mouse analyses; X.C., J.Y., and H.W. wrote the paper.

## Competing interests

The authors declare no competing interests.
