## [Peer Review File · Nature Communications]

REVIEWER COMMENTS

Reviewer #1 (Remarks to the Author):

Chen's work explores the expression, function and mechanism of action of Dhx36 in SC. Although the link between Dhx36 and G-quadruplex has been demonstrated in other studies, here this mechanistic link is shown to impact skeletal muscle stem cell regenerative functions. The role of Dhx36 bound to G4s in SC is new and original but needs to be demonstrated more thoroughly.

1. ".In particular, whether Dhx36 has a regulatory role in the mRNA metabolism of stem "cells has never been addressed." Does the effect of dhx36 on muscle stem cell only depend solely on Gnai2? The authors conclude with "We envision that many other targets, similar to Gnai2, exist in stem cells that mediate the pleiotropic Dhx36 functions, which calls for future intensive investigations." but this should have been addressed here, not in future studies.

2. "To dissect how Dhx36 functions in SCs and muscle regeneration, we...". I will add "whether" because the link has not been established before

3. "Of note, previous studies showed that constitutive activation of Gnai2 resulted in "muscle hypertrophy and accelerated injury-induced muscle regeneration while knocking-out Gnai2 reduced muscle size and impaired muscle regeneration [38, 39], which to some degree phenocopies the impaired muscle regeneration in Dhx36 mutant mice, suggesting that Gnai2 is a likely target of Dhx36." It would be important to show that Gnai2 expression (protein but not mRNA) is increased similarly to dhx36 during SC activation and proliferation (as shown for dhx36 in Fig. 1).

4. Fig. 1C: the increase at SC48h is not evident considering the loading control

5. "As expected, Dhx36 mRNAs were highly enriched at day 3 (3 fold compared to uninjured) and started to decline by day 7 (Fig. 1D)." Is the difference between 3 and 7 days treatment significant?

6. "To further examine its expression during the myogenic differentiation process in vitro, Dhx36 protein levels were also down-regulated when proliferating C2C12 mouse myoblasts were cultured in differentiating medium (DM), particularly at advanced differentiation stages (Suppl. Fig. 1B)." This sentence needs to be revised

7. Sup Fig 1B: not convincing, tubulin band is overexposed

8. "we found that Dhx36 protein was exclusively present in the cytoplasm of activated SCs associated to visible puncta". What are these puncta? Is this specific localization associated to translation regulatory sites? It would be important to discuss this results considering the Sauer manuscript.

9. It would be helpful to indicate the axes labels in all the figures of the manuscript and give information that will help in an easier understanding of the figure (eg Fig 4F: Y axis missing; Fig 4 K,L,M indicate the mRNA region, etc)

10. "total mRNAs from the cell lysates and the fraction of mRNAs associated with polysomes" . Did they mean total mRNAs from the cytoplasmic fraction?

11. To further elucidate the TE changes caused by direct Dhx36 binding, the above data were intercepted with Dhx36 binding targets from the CLIP-seq, uncovering a much higher number (76 vs 38) of TE-down (≤ 0.66 fold) vs TE-up (≥ 1.5 fold) transcripts with Dhx36 binding (Fig. 5D and

Suppl. Table S2), thus suggesting a promoting effect of Dhx36 on global mRNA translation. The conclusion on "global translation" should be toned down, also in agreement with their conclusion on polysomal profiles (Fig 5B)

12. Furthermore, the rG4 structures appeared again dominated by bulges and two quartets. This signature could be experimentally validated. The Gnai2 G4 did not appear to contain bulges.

13. Supporting that this new translational effect is largely related to 5'UTR binding, we found that Dhx36 binding to 5'UTRs conferred a stronger effect on TE change (a total of 73 transcripts with TE changes, Fig. 5G) as compared to bindings within 3'UTRs (19) and CDS (46) (Fig. 5H-I). The number of mRNAs interacting with Dhx36 and targeted for translational regulation seems to be quite small. It would be important to comment on this result in relation to the work of Sauer, Murat and Herviou.

14. The manuscript would benefit from moving some figures to supplementary section (eg 7A, 7C, 7E, 7G, 7H)

15. Fig 7: It would be important to fully demonstrate that Dhx36 mediated translational regulation of Gnai2 depends on G4s. This point could be addressed by using PDS or cPDS in the analysis of RNA-prot interactions, Gnai2 expression and functional consequence in SC. In the same line, Fig 7K should be repeated in absence of Dhx36. Also, it would be important to demonstrate in cellulo RG4 unfolding following Dhx36 increased expression (Fig. 1).

16. Fig 7: To fully support the regulatory model in which Dhx36 is increased during SC activation/proliferation, the authors should validate the results by overexpressing Dhx36

Reviewer #2 (Remarks to the Author):

Chen et al. investigated the role of Dhx36 in muscle stem cell behaviors. The data clearly indicate the critical roles of Dhx36 in muscle regeneration by regulating muscle stem cell proliferation. This study also showed that Dhx36-5'UTR G4 interaction functions to facilitate mRNA translation. Lastly, among the Dhx36-associated targets, the authors investigated the role of Gnai2 in muscle stem cell proliferation. The presented data are convincing, and experiments are well-performed. Please respond to following concerns raised by the Reviewer.

1. Dhx36 tends to reduce RNA abundance, but Dhx36 promotes TE. Although the authors speculate that reduced RNA abundance may result from retarded translation, it is a little confusing if considering total protein amount. Is there any data indicating this speculation? In addition, please describe whether Dhx36 loss affect the amount of Gnai2.

2. Page 7; 'we found that Dhx36 protein was exclusively present in the cytoplasm of activated SCs associated to visible puncta (Fig. 1E).'

Dhx36 regulates DNA, but the results of Fig.1E shows that all Dhx36 proteins localized in cytoplasm. Please discuss about the specific localization of Dhx36 in muscle stem cells. In addition, the result of SCT0 should be included in the Figure.

3. Page 10-11

Figure 3 and S. Fig. 2 showed that the loss of Dhx36 has no impact on cell cycle re-entry and MyoD expression in MuSCs. However, the authors mention that Dhx36 is critical for MuSC

activation/proliferation. It is unclear the definition of 'activation' here.

4. More mice number should be added in S.Fig. 1C (N=3 for Ctrl group and 4 for cKO group). At least, it will possible to add N=1 for both Ctrl and cKO group without more experiments, because N=4 for Ctrl group and 5 for cKO group are used in S.Fig.1D.

5. Fig.4F

If possible, please use same color for indicating the RNA-binding site.

6. Fig.4E and 4G

Please add more information about the analyzed genes in Fig. 4 E and G.

7. Fig.5 G-H, Fig5K-M

Are there any relevance between Fig.5G-I and Fig.5K-M? Reviewer thought that up or down TE genes/sites in Fig.5G-I were further analyzed in Fig.5K-M. However, the number was not matched in some of them.

8. Fig. 8

It is unclear how Gnai2 regulates MuSC proliferation. Please add some descriptions in discussion.

Point by point responses

Reviewer #1 (Remarks to the Author):

Chen's work explores the expression, function and mechanism of action of Dhx36 in SC. Although the link between Dhx36 and G-quadruplex has been demonstrated in other studies, here this mechanistic link is shown to impact skeletal muscle stem cell regenerative functions. The role of Dhx36 bound to G4s in SC is new and original but needs to be demonstrated more thoroughly.

1.1. “..In particular, whether Dhx36 has a regulatory role in the mRNA metabolism of stem "cells has never been addressed.” Does the effect of dhx36 on muscle stem cell only depend solely on Gnai2? The authors conclude with “We envision that many other targets, similar to Gnai2, exist in stem cells that mediate the pleiotropic Dhx36 functions, which calls for future intensive investigations.” but this should have been addressed here, not in future studies.

We thank the reviewer for the comment. As a highly potent RBP regulator, DHX36 has many targets in any given cell. In our study, a total of 3662 binding targets on 1262 transcripts were identified in myoblasts by CLIP-seq profiling. Therefore, we believe it is very possible that DHX36 function in muscle stem cells (satellite cells, SCs) can be mediated by more than one target. Indeed, 61 out of 73 targets showed that translational efficiency (TE) was downregulated upon DHX36 loss (Fig. 5d); among these, 7 were related to cell proliferation regulation (YAP1, DLG1, ADAM10, SSBP3, Gnai2, DAZAP1, Gnb1). However, due to the limitation of resources and time, we focused our study on an in-depth dissection of regulatory mechanisms using the best candidate target: *Gnai2*. We demonstrated that it is a major mediator of DHX36 function in promoting myoblast proliferation. It will be interesting to elucidate how these other targets may mediate DHX36 function through other diversified mechanisms, which are sure to generate exciting findings for separate papers; however, we feel it is not only out of scope for this manuscript but also may distract from our main message. Nevertheless, we have now revised the text on page 19 to address the reviewer's point and added a new Supplementary Fig. 5d-e to illustrate other possible targets. We hope the reviewer will find the answer acceptable.

1.2. “To dissect how Dhx36 functions in SCs and muscle regeneration, we...”. I will add “whether” because the link has not been established before.

Thank you for the comment. We agree that “whether” is better, as the functional roles had never been studied in skeletal muscle SCs. The text has been revised as suggested on page 7.

1.3. “Of note, previous studies showed that constitutive activation of Gnai2 resulted in “muscle hypertrophy and accelerated injury-induced muscle regeneration while knocking-out Gnai2 reduced muscle size and impaired muscle regeneration [38, 39], which to some degree phenocopies the impaired muscle regeneration in Dhx36 mutant mice, suggesting that Gnai2 is a likely target of Dhx36.” It would be important to show that Gnai2 expression (protein but not mRNA) is increased similarly to dhx36 during SC activation and proliferation (as shown for dhx36 in Fig. 1).

We thank the reviewer for the comment. We have now conducted Western blot of *Gnai2* in SCs and indeed *Gnai2* protein expression was increased upon activation and maintained during SC proliferation, which was similar with the expression dynamics of DHX36 protein. The newly added data can be found in Fig. 7c and page 19 of the text.

1.4. Fig. 1C: the increase at SC48h is not evident considering the loading control.

We agree that the increase of DHX36 expression at 48 h in SCs in the original Fig. 1c was not evident, mainly due to technical reasons (unequal loading of the gel). We have now repeated the Western blot with equal

loadings. As shown in the new Fig. 1c, we now conclude that DHX36 was indeed highly induced upon activation (in SC at 24 h), but no further increase was observed at 48 h. The text has now been revised on page 7.

1.5. *“As expected, Dhx36 mRNAs were highly enriched at day 3 (3 fold compared to uninjured) and started to decline by day 7 (Fig. 1D).” Is the difference between 3 and 7 days treatment significant?*

Thank you for the comment. We have made the calculation and found no significant difference. We believe this is because the expression levels were examined in SCs that had been FACS isolated post-injury, meaning that they were early differentiating myoblasts but not fully differentiated myotubes. As shown in Supplementary Fig. 1b, DHX36 is sharply decreased only in late but not early differentiating stage. To better illustrate its decrease at the differentiation stage *in vivo*, we have now repeated the experiment using whole muscle homogenates post injury and found that the expression of Dhx36 mRNAs was indeed highly induced upon injury and reached the peak at day 3 (6.6-fold compared to day 0). And expectedly, a significant decline was observed afterwards; at day 7 post injury DHX36 was decreased by 8.8-fold compared to the day 3 level. We have now included these newly acquired data in the revised Fig. 1d and have revised the text on page 7.

1.6. *“To further examine its expression during the myogenic differentiation process in vitro, Dhx36 protein levels were also down-regulated when proliferating C2C12 mouse myoblasts were cultured in differentiating medium (DM), particularly at advanced differentiation stages (Suppl. Fig. 1B).” This sentence needs to be revised*

Thank you for the comment. We apologize for our unclear sentence, which we have now revised it to read: "Additionally, in C2C12 mouse myoblasts cultured in differentiating medium (DM), the DHX36 protein levels also decreased during the differentiation course, and particularly at late differentiation stages (72 h and 120 h), further confirming that it is downregulated during myogenic differentiation (Supplementary Fig. 1b)." on page 7. We hope this is clearer.

“”

1.7. *Sup Fig 1B: not convincing, tubulin band is overexposed*

We have now repeated the experiments using the GAPDH protein as a normalization control and have replaced the original Supplementary Fig. 1d.

1.8. *“we found that Dhx36 protein was exclusively present in the cytoplasm of activated SCs associated to visible puncta”. What are these puncta? Is this specific localization associated to translation regulatory sites? It would be important to discuss this results considering the Sauer manuscript.*

Thanks for the suggestion. We agree it will be important to discuss the nature of the puncta. Sauer et al.[1] demonstrate that in human HEK293 cells, DHX36 was enriched in the ribosome-free and monosomal fractions, and exhibited a similar pattern as EIF4A1, which is the subunit of translational initiation complex. Moreover, Herviou et al.[2] also demonstrated that the sedimentation pattern of DHX36 was highly similar to that of EIF4A in U251 glioblastoma cells. However, co-IP in U87 cells showed no direct interactions between DHX36 and EIF4A. In our cells, we performed co-IP and did not detect evident interactions between DHX36 and EIF3B, a component of translation initiation complex (data not shown). We think that more exhaustive experiments will be needed to test if DHX36 localizes to translational regulatory sites. Nevertheless, in a separate investigation, we discovered that DHX36 can form non-membrane organelles through liquid-liquid phase separation (LLPS). As shown below **(a)**, discrete puncta were observed in the cytoplasm when EGFP-fused DHX36 protein were overexpressed in 293T cells. To further validate the formation of membrane-less bodies, fluorescence recovery after photobleaching (FRAP) assay was performed, which revealed that fluorescence recovered after bleaching, at the indicated rate (see below, **b**). These preliminary results indicate that DHX36 has the capacity

of driving LLPS formation in cells. We thus speculate that the observed puncta in myoblast cells represent sites of DHX36 initiated LLPS, which is a very exciting scenario that we are currently investigating. We have revised the manuscript on page 24 to include the above points.

1.9. *It would be helpful to indicate the axes labels in all the figures of the manuscript and give information that will help in an easier understanding of the figure (eg Fig 4F: Y axis missing; Fig 4 K,L,M indicate the mRNA region, etc).*

Thanks for the comments. We apologize for the unclear labeling and have now revised the figures as suggested.

1.10. *“total mRNAs from the cell lysates and the fraction of mRNAs associated with polysomes” . Did they mean total mRNAs from the cytoplasmic fraction?*

We apologize for the unclear writing. Yes, this indicates total mRNAs from the cytoplasmic fraction and the text has been revised on page 15.

1.11. *To further elucidate the TE changes caused by direct Dhx36 binding, the above data were intercepted with Dhx36 binding targets from the CLIP-seq, uncovering a much higher number (76 vs 38) of TE-down (≤ 0.66 fold) vs TE-up (≥ 1.5 fold) transcripts with Dhx36 binding (Fig. 5D and Suppl. Table S2), thus suggesting a promoting effect of Dhx36 on global mRNA translation. The conclusion on “global translation” should be toned down, also in agreement with their conclusion on polysomal profiles (Fig 5B).*

Thanks for the comment. We agree that we need to tone down our conclusion, as only around 100 target mRNAs were affected upon Dhx36 loss. The text on page 15 has now been revised to: “To further elucidate the TE changes caused by direct DHX36 binding, the above data were intercepted with DHX36 binding targets from the CLIP-seq, which uncovered a much higher number (76 vs 38) of TE-down vs TE-up transcripts with DHX36 binding (Fig. 5d, Supplementary Table 2). These results suggested that DHX36 selectively promotes the translation of a portion of target mRNAs.”

1.12. *Furthermore, the rG4 structures appeared again dominated by bulges and two quartets. This signature could be experimentally validated. The Gnai2 G4 did not appear to contain bulges.*

We agree that experimental validation would strengthen our finding. To this end, we have now selected five of the potential mRNAs harboring bulge or two quartet rG4 structures at the DHX36 bound 5'-UTRs (listed in Supplementary Fig. 4c), including Cd82, Hnrnp1, Mknk2, Nid2 and Pja2 and performed G4 ligand enhanced fluorescence spectroscopy, CD spectroscopy and UV melting assays. As shown in Supplementary Fig. 4a-d (page 12-13 of the revised text), the results indeed confirmed the formation of rG4 structures at the predicted

sites. However, our labs are not equipped to perform high resolution structural assay to determine the exact subtype of the rG4s; in any case, we feel this is out of scope for our main message. We hope this is acceptable for the reviewer and remain open to additional suggestions or comments on this point.

1.13. *Supporting that this new translational effect is largely related to 5'UTR binding, we found that Dhx36 binding to 5'UTRs conferred a stronger effect on TE change (a total of 73 transcripts with TE changes, Fig. 5G) as compared to bindings within 3'UTRs (19) and CDS (46) (Fig. 5H-I). The number of mRNAs interacting with Dhx36 and targeted for translational regulation seems to be quite small. It would be important to comment on this result in relation to the work of Sauer, Murat and Herviou.*

Thank you for this insightful comment. We agree that the number of mRNAs interacting with DHX36 and targeted for translational regulation is unexpectedly small (381 targets were found in Sauer et al.'s study). We reason that this may stem from the relatively small number of targets identified by our CLIP-seq. A total of 3032 peaks corresponding to 1262 genes were retrieved, which is much smaller than the number of targets identified by Sauer et al.[1]. In Sauer's study, CLIP-seq was conducted on ectopically expressed FLAG/HA tagged Dhx36 protein (FH- DHX36) or catalytically dead DHX36 mutant (FH- DHX36-E335A), and a total of 19,585 and 67,660 binding sites were identified, corresponding to more than 4500 mRNAs. In our study, to better understand the physiological role of DHX36, the CLIP-seq was performed on endogenous Dhx36, which may have caused the difference in the number of the identified sites/targets. In addition, different cells (C2C12 vs HEK293 cells) were used in the studies, and the number of regulatory targets could be very likely to be cell-type specific.

As for the study from Murat et al.[3], it was shown that TE of 1026 or 2119 transcripts were significantly affected by depletion of DHX36 or Dhx9, respectively. However, no CLIP-seq data were available for DHX36 to identify the exact regulatory targets. CLIP-seq of Dhx9 was conducted, but the exact number of direct targets with significant TE changes was not shown. Considering that only 392 of the identified binding peaks (not transcripts) were at the 5'-UTR region, we speculate that the number of mRNAs interacting with Dhx9 and targeted for translational regulation is comparable with ours.

Lastly, in the recent study by Herviou et al.[2], polysome profiling followed by qRT-PCR was adopted to examine the cooperative effect of HNRNP H/F and DHX36 on target translation; as only five mRNAs rather than the whole transcriptome were analyzed, it lacked a global view of HNRNP H/F and Dhx36 on translational regulation.

We have revised the text on page 24 to include the above points.

1.14. *The manuscript would benefit from moving some figures to supplementary section (eg 7A, 7C, 7E, 7G, 7H).*

Thank you for the comment. We have now moved some of the mentioned figures (Fig. 7c, 7e, 7g, 7h, 7l) to the supplementary section as suggested.

1.15. *Fig 7: It would be important to fully demonstrate that Dhx36 mediated translational regulation of Gnai2 depends on G4s. This point could be addressed by using PDS or cPDS in the analysis of RNA-prot interactions, Gnai2 expression and functional consequence in SC. In the same line, Fig 7K should be repeated in absence of Dhx36. Also, it would be important to demonstrate in cellulo RG4 unfolding following Dhx36 increased expression (Fig. 1).*

Thank you for the critical comment. We agree that it would be important to fully demonstrate that DHX36-mediated translational regulation of Gnai2 depends on G4s. As suggested, we have now treated activated SCs with cPDS and found that Gnai2 protein expression was significantly downregulated by the treatment, while

mRNA levels increased (Fig. 7l). To confirm the result, we also conducted the same assay with proliferating C2C12 myoblast cells; this showed that the Gnai2 protein level decreased significantly without evident mRNA changes (Suppl. Fig. 8f), suggesting that the Gnai2 protein level is indeed regulated by the rG4 structure. As suggested, cPDS treatment was also used in the RNA immunoprecipitation assay (RIP); we found the treatment indeed significantly increased DHX36/Gnai2 interaction (Fig. 7h). In addition, we also demonstrate that SC proliferation was significantly decreased by cPDS treatment (Suppl. Fig. 8g).

As for Fig. 7k, the original experiment was performed in the absence of DHX36. We apologize for the misleading writing. To strengthen the conclusion, we have now conducted additional reporter assays in *Dhx36* KO cells and found that the EGFP protein levels of all three reporters (FL, rG4#1, rG4#2) were dramatically downregulated in *Dhx36* KO cells, while the mRNA levels remained unaltered. These newly added data are in Fig. 7q–s.

To examine the *in cellulo* rG4 formation, we have now stained SCs with the rG4-specific fluorescent ligand QUMA-1[4]. Interestingly, in contrast to the increasing DHX36 level during SC activation, we did not observe corresponding decrease of rG4 signals in ASC at 24 h as compared to SC_{T8} cells (Suppl. Fig. 9). We reason that since SC activation involves dynamic changes of gene regulation at many levels, and since DHX36 may not be the only rG4 regulator during the process, it is possible not to detect a negative correlation between *in cellulo* rG4 levels and DHX36 expression. Nevertheless, in activating SCs, we did detect significantly increased QUMA-1 signals in *Dhx36* iKO vs control SCs (Fig. 7m), confirming a DHX36-dependent unfolding of rG4 in activated SC.

Altogether, we believe the newly acquired data have strengthened the finding that DHX36-mediated translational regulation of Gnai2 depends on the rG4 structures formed in the 5'-UTR. Accordingly, we have revised the text on pages 20 and 21.

1.16. Fig 7: To fully support the regulatory model in which *Dhx36* is increased during SC activation/proliferation, the authors should validate the results by overexpressing *Dhx36*.

Thanks for the comment. We have now performed *Dhx36* overexpression. As expected, the overexpression significantly increased SC proliferation and Gnai2 protein level, supporting the regulatory model that *Dhx36* promoted Gnai2 expression and SC proliferation. The newly acquired data can be found in Fig. 8g-h, Suppl. Fig. 10 and page 21-22 of the revised text.

Reviewer #2 (Remarks to the Author):

*Chen et al. investigated the role of *Dhx36* in muscle stem cell behaviors. The data clearly indicate the critical roles of *Dhx36* in muscle regeneration by regulating muscle stem cell proliferation. This study also showed that *Dhx36*-5'UTR G4 interaction functions to facilitate mRNA translation. Lastly, among the *Dhx36*-associated targets, the authors investigated the role of *Gnai2* in muscle stem cell proliferation. **The presented data are convincing, and experiments are well-performed.** Please respond to following concerns raised by the Reviewer.*

2.1. *Dhx36* tends to reduce RNA abundance, but *Dhx36* promotes TE. Although the authors speculate that reduced RNA abundance may result from retarded translation, it is a little confusing if considering total protein amount. Is there any data indicating this speculation? In addition, please describe whether *Dhx36* loss affect the amount of *Gnai2*.

Thanks for the critical comment. In the analysis (Fig. 6b) we found a high percentage of up-regulated mRNAs possessing 5'UTR binding on rG4 sites, leading to the speculation that *Dhx36* loss increases transcript abundance, which could result from a feedback reaction to the retarded translation[5, 6]. In addition, the total amount of proteins was not significantly changed as shown in Fig. 5b. Concerning the Gnai2 protein level upon

Dhx36 deletion, as shown in Fig. 7i-j, Dhx36 loss in both SC and C2C12 myoblasts resulted in dramatically decreased Gnai2 protein levels and slightly increased mRNA level. These data indicate that the Gnai2 translation was regulated by DHX36, and that the retarded translation efficiency caused by Dhx36 loss cannot be compensated by an increased mRNA level. We think the increase of Gnai2 mRNA expression is more likely to be a feedback effect of decreased Gnai2 protein translation. We have added the discussion on page 25 of the revised text.

2.2. *Page 7; ‘we found that Dhx36 protein was exclusively present in the cytoplasm of activated SCs associated to visible puncta (Fig. 1E).’ Dhx36 regulates DNA, but the results of Fig. 1E shows that all Dhx36 proteins localized in cytoplasm. Please discuss about the specific localization of Dhx36 in muscle stem cells. In addition, the result of SCT0 should be included in the Figure.*

Thanks for the comment. Indeed, DHX36 can regulate both G4 DNA and RNA structures. We believe that its dominant localization in cytoplasm of ASCs suggests that it mainly functions as a rG4 regulator. This has been added on page 8 of the revised manuscript. As for the puncta staining pattern in Fig. 1e, the Reviewer 1 has also raised a similar comment. Please see above our answer to her/his comment 1. 8 (above).

As suggested, we have now stained DHX36 in the fixed quiescent SCs (SC_{T0}) and found the signal was very weak (Fig. 1e), which was consistent with the low level of DHX36 in SC_{T0} (Fig. 1b).

2. 3. *Page 10-11, Figure 3 and S. Fig. 2 showed that the loss of Dhx36 has no impact on cell cycle re-entry and MyoD expression in MuSCs. However, the authors mention that Dhx36 is critical for MuSC activation/proliferation. It is unclear the definition of ‘activation’ here.*

Thanks for the critical comment. We apologize for the confusing use of activation vs proliferation. SC activation occurs when SCs exit their quiescent state and re-enter cell cycle followed by cell mitosis and proliferation. Our data in Fig. 3 and Suppl. Fig. 2 showed no significant difference of MyoD or Ki67 protein expression in control or Dhx36 iKO SCs, therefore we conclude now that SC activation is not impacted by Dhx36 loss. We have now revised the text as “proliferation/cell cycle progression” on pages 11.

2. 4. *More mice number should be added in S.Fig. 1C (N=3 for Ctrl group and 4 for cKO group). At least, it will possible to add N=1 for both Ctrl and cKO group without more experiments, because N=4 for Ctrl group and 5 for cKO group are used in S.Fig. 1D.*

We have now included more mice for both groups (n=6 for control, and n=7 for cKO) as shown in the revised Suppl. Fig. 1c and 1d. The text has also been revised on page 8.

2. 5. *Fig.4F. If possible, please use same color for indicating the the RNA-binding site.*

We have revised Figs. 4e-g to use the same color for RNA-binding site.

2. 6. *Fig.4E and 4G, please add more information about the analyzed genes in Fig. 4 E and G.*

Thanks for the comments. We have now integrated the detailed gene list in spreadsheets “mRNAs with different sites” and “co-existing binding information” of Suppl. Table S1.

2. 7. *Fig.5 G-H, Fig5K-M, Are there any relevance between Fig.5G-I and Fig.5K-M? Reviewer thought that up or down TE genes/sites in Fig.5G-I were further analyzed in Fig.5K-M. However, the number was not matched in some of them.*

We apologize for being unclear. Indeed, the two sets of figures are relevant. Figs. 5k-m are to examine the rG4 structure constitution of the binding sites on the mRNAs from Fig. 5g-i. Since multiple binding sites can exist on one mRNA (Fig. 5g), the numbers differ in the two sets of figures. We have now revised the figure labels and the text on page 16 to clarify this point.

2. 8. Fig. 8, It is unclear how *Gnai2* regulates MuSC proliferation. Please add some descriptions in discussion.

We apologize for our unclear description of *Gnai2* function and mechanisms in regulating SC proliferation. This has been well demonstrated by a prior study by Minetti et al.[7]. We have now revised the text on page 26.

Reference

1. Sauer, M., et al., *DHX36 prevents the accumulation of translationally inactive mRNAs with G4-structures in untranslated regions*. Nat Commun, 2019. **10**(1): p. 2421.
2. Herviou, P., et al., *hnRNP H/F drive RNA G-quadruplex-mediated translation linked to genomic instability and therapy resistance in glioblastoma*. Nat Commun, 2020. **11**(1): p. 2661.
3. Murat, P., et al., *RNA G-quadruplexes at upstream open reading frames cause DHX36- and DHX9-dependent translation of human mRNAs*. Genome Biol, 2018. **19**(1): p. 229.
4. Chen, X.C., et al., *Tracking the Dynamic Folding and Unfolding of RNA G-Quadruplexes in Live Cells*. Angew Chem Int Ed Engl, 2018. **57**(17): p. 4702-4706.
5. Devany, E., et al., *Positive and negative feedback loops in the p53 and mRNA 3' processing pathways*. Proc Natl Acad Sci U S A, 2013. **110**(9): p. 3351-6.
6. Ji, N., et al., *Feedback control of gene expression variability in the Caenorhabditis elegans Wnt pathway*. Cell, 2013. **155**(4): p. 869-80.
7. Minetti, G.C., et al., *Galphai2 signaling is required for skeletal muscle growth, regeneration, and satellite cell proliferation and differentiation*. Mol Cell Biol, 2014. **34**(4): p. 619-30.

REVIEWERS' COMMENTS

Reviewer #1 (Remarks to the Author):

Chen and colleagues provided compelling answers to my questions, supported by insightful explanations and/or the addition of solid complementary experiences. I recommend the publication of their article in nat comm

Reviewer #2 (Remarks to the Author):

The authors addressed all concerns raised by this Reviewer. Please correct following minor issues.

1. Related to original comment #3

Please check the term 'activation' in the following description. 'Proliferation' or 'Re-entry to cell cycle' is adequate because increased rG4 signals were detected between ASC-24h and -48hr.

Page 21; Interestingly, despite increasing DHX36 levels, we did not observe corresponding decrease of rG4 signals in ASC 24 h vs SC-T8 cells (Supplementary Fig. 9a, b), probably indicating that DHX36 is not the sole rG4 regulator in SCs activation.

2. Reviewer felt that the following description is an overstatement as the phenotype is not so severe.

Page 23; loss of Dhx36 completely blunted muscle regeneration with no compensation by other helicases;

Reviewer #2 (Remarks to the Author):

The authors addressed all concerns raised by this Reviewer. Please correct following minor issues.

1. Related to original comment #3

Please check the term 'activation' in the following description. 'Proliferation' or 'Re-entry to cell cycle' is adequate because increased rG4 signals were detected between ASC-24h and -48hr.

Page 21; Interestingly, despite increasing DHX36 levels, we did not observe corresponding decrease of rG4 signals in ASC 24 h vs SC-T8 cells (Supplementary Fig. 9a, b), probably indicating that DHX36 is not the sole rG4 regulator in SCs activation.

Thanks for the comment. We have now revised the text as suggested on page 21.

2. Reviewer felt that the following description is an overstatement as the phenotype is not so severe.

Page 23; loss of Dhx36 completely blunted muscle regeneration with no compensation by other helicases;

Thanks and we have now revised the description on page 23 as "loss of Dhx36 dramatically impaired muscle regeneration with no compensation by other rG4-resolving helicases".